



# Soil $CO_2$ efflux from two mountain forests in the Eastern Himalayas Bhutan: components and controls

Norbu Wangdi[1,2], Mani Prasad Nirola[1,4], Mathias Mayer[1], Norbu Zangmo[2], Karma Orong[2], Iftekhar Uddin Ahmed[1], Andras Darabant[1], Robert Jandl [3], Georg Gratzer[1], Andreas Schindlbacher[3]

[1] Institute of Forest Ecology, University of Natural Resources and Life Sciences, 1180 Peter Jordan Strasse, Vienna, Austria

[2] Ugyen Wangchuck Institute for Conservation and Environmental Research, Department of Forests and Park Services Lamai Goempa, Bumthang, Bhutan

[3] Federal Research and Training Centre for Forests, Natural Hazards and Landscape – BFW, A-1131 Vienna, Austria

[4] National Biodiversity Center, Ministry of Agriculture and Forests, Thimphu, Bhutan

*Correspondence to*: Norbu Wangdi (norwangs@gmail.com)

**Abstract**

The biogeochemistry of mountain forests in the Hindu Kush-Himalaya range is poorly studied although climate change is expected to disproportionally affect the region. We measured the soil $CO_2$ efflux (Rs) at a high elevation (3260 m) coniferous, and a lower elevation (2460 m) broadleaved forest in Bhutan, eastern Himalayas, during 2014 and 2015. Both sites experienced typical monsoon weather (cold-dry winters, warm-wet summers) during the study. Trenching was applied to estimate the contribution of autotrophic (Ra) and heterotrophic (Rh) soil respiration. The temperature ($Q_{10}$) and the moisture sensitivities of Rh were determined under controlled laboratory conditions and were used to model Rh in the field. The higher elevation coniferous forest had a higher standing tree stock, reflected in higher soil C stocks and basal soil respiration ($R_{10}$). Rs was similar between the two forests (2015: $14.5 \pm 1.2$ t C yr$^{-1}$ broadleaved; $12.8 \pm 1.0$ t C yr$^{-1}$ coniferous). Modelled annual contribution of Ra was ~ 45% at both forests with a low autotrophic contribution during winter and high contribution during the monsoon season. Ra, estimated from trenching, was lower and highly variable, indicating that trenching poorly performed at these forests/soils. Rs neatly followed the annual course of soil temperature (field $Q_{10}$ between 4 and 5) at both sites. Co-variation between soil temperature and moisture likely was the main cause for the high $Q_{10}$ obtained from field Rs. Temperature sensitivity of Rh was lower ($Q_{10}$ ~ 2.3 at both sites). Under the preceding weather conditions, a simple temperature-driven model was able to explain more than 90% of the temporal variation in Rs. To predict and understand how Rs responds to infrequently occurring extreme climate conditions such as monsoon failures, however, longer Rs time series are required for a better integration of interactions between soil temperature, moisture, Ra and Rh.

**Keywords:** Himalaya, soil $CO_2$ efflux, autotrophic soil respiration, heterotrophic soil respiration, incubation, temperature sensitivity, moisture sensitivity





## 1 Introduction

Carbon dioxide ($CO_2$) efflux from soil (= soil respiration; Rs) is one of the major fluxes in the global C cycle, affects atmospheric $CO_2$ concentrations and feeds back on global climate change (Schlesinger and Andrews, 2000; Reichstein et al., 2003; Hashimoto et al., 2015). Counteracting to C uptake via photosynthesis, Rs primarily determines whether forest ecosystems serve as C sink or source to the atmosphere (Dixon et al., 1994; Schlesinger and Andrews, 2000; Bolstad et al., 2004). The current function of forests as global C sink (Janssens et al., 2003; Stocker, 2014) could weaken or even turn into the opposite if climate change disproportionally accelerates respiratory processes such as Rs (Cox et al., 2000). Rs consists of an autotrophic component (Ra; root and rhizosphere respiration), which is closely linked to C gain by photosynthesis and a heterotrophic component (Rh), which is the respiratory product of soil organic matter (SOM) decomposition. While the source of Ra is recently assimilated $CO_2$, Rh can release stored soil C to the atmosphere. For better prediction of the response of forest C cycling to climate change, it is crucial to understand how Rs and its components are affected by changing environmental parameters such as temperature and moisture (Davidson and Janssens, 2006). Rates and climate sensitivity of Rs, Ra and Rh can vary among forest ecosystem type and climatic region (Hashimoto et al., 2015). So far, research has primarily focused on the temperate and boreal areas of the northern hemisphere and remote forested areas are still largely uninvestigated (Bond-Lamberty and Thomson, 2010). The Hindu Kush-Himalaya range represents a region, where research on forest biogeochemistry is gaining momentum (Ohsawa, 1991; Wangda and Ohsawa, 2006a; Pandey et al., 2010; Sharma et al., 2010b; Verma et al., 2012; Sundarapandian and Dar, 2013; Dorji et al., 2014b; Tashi et al., 2016). It extends over 4.3 million $km^2$ across eight countries with an average forest cover of approximately 20 % (Schild, 2008), ranging from lowland tropical forest to high altitudinal forests up to ~ 4900 m (Schickhoff, 2005; Liang et al., 2016). Situated in the south-eastern range of the Himalayas, Bhutan shows a forest cover of 70 % (DoFPS, 2011). Most forests in Bhutan are natural old growth (Ohsawa, 1987), store high amounts of C in biomass and soil (Sharma and Rai, 2007; Dorji et al., 2014a) and serve as an important regional C sink (FAO, 2010). As climate change is expected to intensify in the Himalaya region (Xu et al., 2009; Tsering et al., 2010; Singh, 2011; Shrestha et al., 2012; Xu and Grumbine, 2014), the effects on forest C cycling could have implications not only regionally, but also on a global scale.

With the objective of a better understanding of soil C cycling of mountain forest ecosystems of the eastern Himalayas, we studied Rs, its components (Ra, Rh), as well as the effects of environmental drivers such as temperature and moisture at a high altitude cool temperate conifer forest and a lower altitude cool broadleaved forest in Bhutan. We hypothesized that (I) overall rates of Rs were higher at the lower elevation and correspondingly warmer broadleaved forest site. As precipitation was expected to be non-limiting during the growing season (~ monsoon season), we further hypothesized that (II) the seasonal course of Rs was mainly driven by soil temperature. The contribution of Ra was expected to be lowest during the cold and dry winter and to significantly increase during the growing season. We further expected that water logged soil showed decreased Rs during peak monsoon.



## 2 Materials and methods

### 2.1 Site description

Two representative forest ecosystems for the eastern Himalayas (Wikramanayake, 2002), a cool temperate mixed coniferous forest and a cool temperate broadleaved forest, were studied at Thimphu and Wangduephodrang districts, Bhutan. The cool temperate mixed coniferous forest (Grierson and Long, 1983) was situated on a south-east facing slope close to the top of a mountain ridge (elevation 3260 m a.s.l). The cool temperate broadleaved forest was situated on an east facing gentle slope along the same mountain ridge ~ 11 km eastwards (elevation 2640 m a.s.l.). Sites will be referred to as "coniferous forest" and "broadleaved forest" in the further text. The coniferous forest was dominated by *Tsuga dumosa* along with *Picea spinulosa*, *Quercus semecarpifolia*, *Abies densa*, *Acer campbelli* and *Taxus baccata*. The broadleaved forest was dominated by *Quercus lanata* and *Quercus griffithii*. Soils at the coniferous forest were Cambisols. Soils at the broadleaved forest were Luvisols. A detailed site and soil description and the comparison are given in Table 1. The current study was aligned within a larger-scale throughfall manipulation experiment, which consisted of control and temporarily roofed areas within each forest type. For this study, we randomly distributed all our plots within the control areas (~ 1500 m$^2$ each) of the throughfall manipulation experiment.

### 2.2 Field measurements

Basic climate parameters were measured using automatic weather stations located at a distance of approx. one kilometer from the sites at the same elevation. Data was recorded at 15 min intervals on a Decagon-EM50 data logger (Decagon Devices Inc., Pullman, WA, USA). The automatic weather stations recorded precipitation with an ECRN-100 rain gauge (Decagon Devices Inc., Pullman, WA, USA), and air temperature and relative humidity with a VP-3 vapor pressure, temperature and relative humidity sensor (Decagon Devices Inc., Pullman, WA, USA).

Stand and soil inventories were carried out in March and April 2014 at both forest types covering an area of ~ 1500 m$^2$ each. The location, height and the diameter at breast height of all trees having a dbh > 10 cm were assessed. The basal area was calculated for each tree species. Standing volume was estimated based on species specific volume equations developed by Paul Lawmans (1994), Forest Survey of India (1996) and Department. of Forests and Park Services, Bhutan (2005). Aboveground litter-fall was collected monthly (since December 2014) using mesh-traps (n = 10) per site, with an area of 1.0 m$^2$ (100 × 100 cm). Litter was dried at 80 °C and the C content was assumed to be 50 % of the dry weight (de Wit et al., 2006). Soil samples were collected from the 0-10, 10-20 and 10-30 cm mineral soil layers of four locations at both sites in May 2014. Soil samples were sieved (2 mm) and dried (105 °C, 48 h). Soil organic C (SOC) of a grinded (Pulverisette 5, Fritsch, Germany), 0.1 g subsample was measured by means of the dry combustion technique using a CN Analyser (TruSpec® CN, LECO Inc., Michigan, USA). Soil organic C stocks (t ha$^{-1}$) were calculated for each horizon by multiplying the SOC concentration (%) by the bulk density (g cm$^{-3}$) and the depth of the horizon (cm). Samples from the forest floor layer were collected in September 2015 and SOC contents were determined as described above.





Rs was measured regularly in the two forest types (coniferous, broadleaved) once every three weeks, from May 2014
to December 2014 and from April 2015 to December 2015. We randomly set 10 plots (n = 10) at each forest type for
Rs measurements. To cover the within-plot variability, Rs was measured at four positions within each plot (total 40
positions per site). We used a portable infrared gas analyzer (EGM-4, PP-Systems, Amesbury, USA) with an attached
soil respiration chamber (SRC-1, PP-Systems, Amesbury, USA) for Rs measurements. In spring 2015 we installed
four permanent collars (total height 5 cm, 2-3 cm inserted into the soil, diameter 10 cm) at each plot which served as
a base for Rs measurements thereafter. Due to logistic reasons, collars were not available in 2014. During the 2014
season, the SRC-1 chamber was directly placed on the ground surface and slightly pressed into the soil to produce
sufficient sealing during Rs measurements. Rs was estimated by a linear fit to the increasing headspace $CO_2$
concentration over time (chamber closure time 90 seconds). A soil respiration measurement campaign lasted for ~
5 h at each site. Measurement order among plots and collars was fully random to avoid any error from temporal
variations in Rs.
We installed two trenching plots at each site in 2014 to estimate the relative contributions of Ra and Rh. Two
additional trenching plots per site were installed in 2015 to increase replication. Trenches (1.5 x 1.5 m squares) were
dug down ~ 1 m, and all the roots within the trenches were cut. The trenches were sealed with double layered plastic
foil in order to restrict tree root ingrowth. Adjoining to each trenched plot, a corresponding control plot of the same
size was established. Each trenched and control plot hosted three collars for Rs measurements.
Volumetric soil water content (0 - 20 cm soil depth) was measured in the center of each plot using a portable Field
Scout TDR meter (Spectrum Technologies, Inc. Aurora, USA) during Rs measurements. Soil temperature at 5 cm
soil depth was measured with a handheld thermometer probe (Hanna Instruments, Germany) at each Rs measurement
location during 2015. For 2014 only soil temperature records from permanently installed sensors were available. Soil
temperature and soil moisture were measured continuously at soil profile pits (two pits per forest type) with five
combined soil temperature- moisture sensors (TM-5; Decagon Devices, Inc., Pullman, WA, USA) at soil depths
ranging from 5 to 120 cm. Data was recorded at 15 min intervals on Decagon-EM50 data loggers (Decagon Devices,
Inc., Pullman, WA, USA).

**2.3 Laboratory incubation**
About 500 g of mineral soil (0-10 cm depth) and approximately 250 g of forest floor material were sampled at six
random locations (n = 6) at each forest type in mid of September 2015. The mineral soil was homogenized and sieved
(2 mm mesh) and stored at 4 °C, at field moisture for one week prior to transport from Bhutan to Austria for further
processing. Forest floor material was not sieved. Upon arrival in Austria, samples from mineral soil were further
divided into 3 sub-samples to account for potential soil heterogeneity at the individual sampling locations. Samples
were filled into 200 cm$^3$ stainless steel cylinders at approximate field bulk density (~ 0.5 g dry weight cm$^{-3}$ for mineral
soil; ~ 0.1 g dry weight cm$^{-3}$ for forest floor). In total, we incubated 36 sub-samples (cylinders) for mineral soil and
12 sub-samples for the forest floor. Filled cylinders were kept at 4 °C for 5 days for equilibration before incubation.
During incubation, $CO_2$ efflux (= Rh) was measured using a fully automated incubation system. Samples were put





into 2 l containers and their $CO_2$ efflux was determined by a dynamic closed – chamber system (Pumpanen et al.,
2009). For $CO_2$ measurements, containers were sequentially connected to an infrared gas analyzer (SBA-4, PP
Systems International Inc., Amesbury, MA, USA) by means of a tubing system. In the meanwhile, disconnected
containers were ventilated in order to prevent internal $CO_2$ enrichment. $CO_2$ concentration within connected
containers were measured for 6 minutes with a recording interval of 10 sec. Rates of $CO_2$ efflux were calculated from
the headspace $CO_2$ increase during 2 – 6 minutes, after Pumpanen et al. (2009).
Incubation proceeded in two steps. We first incubated at different soil temperatures to assess the temperature
sensitivity of Rh. In a second step, we incubated under different soil moisture contents to assess the sensitivity of Rh
to changes in soil moisture. In addition, we repeated the temperature-runs with wet (140 % Grav.) and dry (30 %
Grav.) soil in order to test for effects of soil moisture on the temperature sensitivity of Rh.
Temperature-incubation started with mineral soil. Soil temperature was increased from 5 °C until 25 °C in 5 °C steps,
with each temperature step lasting for 6 h. At each temperature step, efflux measurements were repeated three times
for each cylinder; to account for a warm up period between the individual temperature steps only a calculated mean
value of the latter two measurements was used for further analysis. After finishing the temperature run, we re-
measured Rh at 10 °C to assess and correct for potential effects of labile C loss during the ~ 30 h incubation. The
forest floor was incubated under the same procedure as mineral soil.
After the temperature-incubations, we set soil moisture of all mineral soil sub-samples to 80 % (gravimetric),
incubated at constant 15 °C for 6 h and measured Rh as described above. Afterwards, the three sub-samples from
each sampling location were split into (i) a sub-sample was kept at constant soil moisture (80 % Grav.), (ii) a sub-
sample was allowed to dry out (60 %, 40 % and 20 % Grav.), and (iii) a sub-sample was progressively watered (100
%, 120 % and 140 % Grav.). In-between repeated incubations (all at 15 °C for 6 h) cylinders were stored at 4 °C.
The whole moisture-incubation procedure lasted for 10 weeks with ~ two-weekly intervals between incubations (time
limiting step was soil drying). We used Rh from the sub-samples which had been kept at constant moisture to correct
for potential decreases in Rh due to a loss in labile C throughout the experiment. After finishing incubations, samples
were dried and actual bulk density, as well as gravimetric and volumetric soil moisture of each sub-sample (cylinder),
was calculated and their total C content was determined (TruSpec® CN, LECO Inc., Michigan, USA). Rh rates were
expressed as µmol $CO_2$ kgC$^{-1}$ s$^{-1}$.
**3 Data analysis**

Effects of forest type on field Rs, soil temperature and moisture were tested by means of repeated-measures ANOVA
with a mixed-effects model structure (Pinheiro and Bates, 2000) separately for each year. The significance level for
this and all other analyses was set at $P < 0.05$. The relationship between field soil temperature and Rs was fitted by
an exponential function (Janssens and Pilegaard, 2003):

$R = R_{10} \times Q_{10}^{(T-10)/10}$            (1)





where R ($\mu$mol $CO_2$ m$^{-2}$ s$^{-1}$) is the measured Rs, T (°C) is the soil temperature at 5 cm depth, $R_{10}$ ($\mu$mol $CO_2$ m$^{-2}$ s$^{-1}$)
is the Rs rate at 10 °C and $Q_{10}$ is the apparent temperature sensitivity (Rs change with a proportional change of 10 °C
in soil temperature). Equation (1) was fitted to the individual plot data for 2014 and 2015 separately for calculating
$Q_{10}$ and $R_{10}$. One sampling date (2015 Jul 16) was excluded from this analysis because heavy rain occurred during
measurements. The relationship between Rs and soil moisture was tested by linear regression analyses. To investigate
the influence of both, soil temperature and soil moisture on Rs, and to account for a possible correlation between
these variables, we used a structural equation modelling approach (Grace, 2006). To consider an exponential relation
between soil temperature and Rs, the latter was log transformed prior to analysis. Data from both years were
incorporated in this analysis.
Cumulative annual Rs of both sites and both years were calculated by linear interpolation of field Rs between
measurement dates of each individual plot (the area beneath the curves in Fig. 1 d). In addition, model parameters of
Eq. (1), together with daily field soil temperatures at 5 cm depth were used to calculate daily field Rs. To account for
a spatial variation in soil temperature, continuously measured data were adjusted to discontinuously measured plot-
data by linear modelling. Cumulative annual Rs rates were calculated by averaging the summed-up daily plot Rs
values. Since full season Rs data and continuous soil temperature data were only available for 2015, annual Rs sums
were only determined for 2015 using this approach.
Average Rh rates from laboratory incubations were calculated for each site, soil horizon (mineral soil, forest floor)
and temperature step (5 – 25 °C), respectively. Equation 1 was fitted to the temperature-incubation data to determine
$Q_{10}$ and $R_{10}$ of Rh. To determine the relationship between soil moisture and Rh, we fitted a Gaussian function to the
moisture-incubation data:

$$R = \beta_0 + \beta_1 \ e^{\left(-0.5 \left(\frac{VWC - \beta_2}{\beta_3}\right)^2\right)}$$  (2)

where R is the measured $CO_2$ efflux from soil samples (Rh), $\beta_i$ are model coefficients and VWC is the volumetric
water content of the samples. This specific function showed the best fit when compared to a set of other response
functions tested.
We followed two approaches to estimate the contribution of Ra and Rh in the field. In a first approach, we used the
trenching data, assuming that the $CO_2$ efflux from the trenched plots represented solely Rh, while the $CO_2$ efflux
from adjacent control plots represented Rs, and accordingly, the difference between trenched and control plot $CO_2$
efflux represented Ra. As trenched plots lack water uptake by tree roots, they were regularly wetter than control plots.
We accounted for that by correcting the soil $CO_2$ efflux for the difference in soil moisture by using Eq. (2).
In a second approach, we applied the response functions of Rh which we had derived during laboratory incubations
together with field soil C stocks and field climate data. This allowed an alternative way to estimate the contribution
of Rh in the field (Gough et al., 2007; Kutsch et al., 2010). Equation (1) and Eq. (2) of each site and soil horizon
were combined:

$$R = f(T) \ f(VWC)$$  (3)




in order to account for both a Rh response to temperature ($f$ (T)) and moisture ($f$ (VWC)). The moisture term in Eq.
(3) was rescaled to relative Rh rates between 0 and 1 (Fig. S1). For that, Eq. (2) was used to predict Rh for a moisture
range between 10 – 70 Vol.%. Predicted Rh rates were then scaled to the asymptote of the curve which represents a
maximum Rh rate. Since no specific moisture response function was obtained for the litter layers, we applied the
same parameters as for mineral soil, combined with $R_{10}$ and $Q_{10}$ parameters for the litter layer. In a second step,
model parameters derived from Eq. (3), together with continuously measured temperature and moisture data from 5
cm soil depth were used to model daily Rh from the litter layer and from the mineral soil in 0 – 10 cm depth
respectively. Model parameters for mineral soils together with continuous measurements of soil temperature and
moisture in 20 cm depth were further used to model daily Rh from the mineral soil in 10 – 30 cm depth. In the last
step, predicted Rh rates ($\mu$mol $CO_2$ kgC$^{-1}$) were multiplied by the C stocks (kg C m$^{-2}$) of the respective soil layer,
which enabled us to upscale Rh to the whole soil profile in the field (Kutsch et al., 2010).





## 4. Results

Air and soil temperatures were ~ 4°C higher at the lower elevation broadleaved forest (Table 1) with a stable trend throughout both study years (Fig. 1). Air temperatures reached a maximum of 29.6 °C and 22.6 °C at the broadleaved and coniferous forest, respectively. Winter air temperatures dropped slightly below freezing at the coniferous forest which showed ephemeral snow cover. Soil temperatures remained above freezing at both sites during the full study period (Fig. 1). Precipitation was higher at the coniferous forest (coniferous 883 mm; broadleaved 688 mm) during precipitation measurements in 2014 (12th Jun – 31st Dec). Annual precipitation in 2015 was similar at both forest types (coniferous 1167 mm, broadleaved 1120 mm). Both sites received the maximum rainfall (60-75 % of annual precipitation) during the peak monsoon months (Jun, Jul and Aug). Soil moisture remained at a similar range (~ 40 Vol. %) at both sites during the summer of 2014, whereas soil moisture was significantly higher at the broadleaved forest during summer 2015 (Fig. 1). During the dry season (Nov – Apr), manually measured soil moisture decreased to < 20 Vol. % at both sites. Continuous soil moisture records indicated accelerated drying at the broadleaved forest (Fig. 1).

Aboveground and below ground C stocks were markedly higher in the coniferous forest (Table 1). Standing volume was 1066 and 464 $m^3$ $ha^{-1}$, at the coniferous and broadleaved forest, respectively. Mineral soil organic C stocks down to 30 cm soil depth were 127 and 91 t $ha^{-1}$ and leaf litter inputs (2015) were 3.5 and 3.4 t C $ha^{-1}$ at the coniferous and broadleaved forest, respectively. Fine root biomass (0-30 cm mineral soil) was lower at the coniferous forest (2.3 t C $ha^{-1}$) when compared to the broadleaved forest (3.2 t C $ha^{-1}$).

Rs did not differ significantly among the two forest types during both years. Rs was generally higher during 2014 (mean Rs broadleaved: $6.7 \pm 1.2$ µmol $CO_2$-C $m^{-2}$ $s^{-1}$, coniferous: $5.6 \pm 0.9$ µmol $CO_2$-C $m^{-2}$ $s^{-1}$) than during 2015 (mean Rs broadleaved: $4.2 \pm 0.7$ µmol $CO_2$-C $m^{-2}$ $s^{-1}$, coniferous: $4.0 \pm 0.6$ µmol $CO_2$-C $m^{-2}$ $s^{-1}$). This difference was not explained by soil climate, nor did we observe any variations in specific above ground dynamics or in litter input during the two study years. We, therefore, attribute the higher Rs rates in 2014 to the methodological differences in Rs measurements. In 2014, Rs was measured without the use of base-collars by inserting the soil respiration chamber into the soil surface. Chamber insertion could have pumped $CO_2$ out of the soil thereby overestimating Rs. We therefore only refer to the cumulative annual Rs from 2015 for further site comparison. Cumulative annual (2015) Rs were $14.3 \pm 0.5$ t C $ha^{-1}$ for broadleaved and $13.0 \pm 0.5$ t C $ha^{-1}$ for the coniferous forest when calculated by linear interpolation between measurement dates. These values were very close to the ones obtained by the modeling approach (Eq. (1)), ($14.5 \pm 1.2$ t C $ha^{-1}$ for broadleaved and $12.8 \pm 1.0$ t C $ha^{-1}$ for coniferous forest) and indicate that a three-week measurement interval is sufficient to explain most of the temporal variability in Rs. Rs showed a higher spatial variability at the coniferous forest (21 - 87 % CV) than at the broadleaved forest (23 - 46 % CV). Between 89 and 96 % of the annual temporal variation in measured Rs could be explained by field soil temperature (Eq. 1, Fig. 2). $Q_{10}$ values of Rs ranged between 3.95 and 5.03 (Fig. 2). Rs showed a weak linear relationship with soil moisture at the broadleaved forest, whereas there was no significant correlation between Rs and soil moisture at the coniferous forest (Fig. 2). For both sites structural equation modelling revealed a strong influence of soil temperature on Rs, but no influence of soil moisture (Fig. 3). At both sites, soil temperature and moisture were strongly correlated with each other during both years (Fig. 3).





Laboratory incubations showed a strong positive, exponential, relationship between soil temperature and Rh (Fig. 2).
Temperature sensitivity of mineral soil Rh was similar among forest types (coniferous $Q_{10}$ = 2.32,
broadleaved $Q_{10}$ = 2.36; Fig. 2) and slightly lower for forest floor material (coniferous $Q_{10}$ = 1.97;
broadleaved $Q_{10}$ = 2.28). $Q_{10}$ values of dry soil (coniferous, 1.59; broadleaved, 1.60) were significantly lower than
$Q_{10}$ from the soil which had been kept at intermediate moisture content ($P < 0.05$, Table 2). $Q_{10}$ values obtained from
dry and wet soil did not differ significantly among the two forest sites (Table 2). Rh and soil moisture showed a
unimodal relationship with highest rates of Rh at intermediate soil moisture (35 - 45 % Vol.) and decreasing rates at
lower and higher moisture levels (Fig. 2). Soil from both forest types responded overall similarly to changes in soil
moisture. Coniferous forest soil showed a slightly sharper decrease in Rh at lower and at higher soil moisture (Fig. 2).
Plots which had been trenched in spring 2014 indicated an average autotrophic contribution of 24 and 30 % at the
coniferous and broadleaved forest during the 2015 season, respectively (Fig. 4). The additional plots, which had been
trenched during spring 2015, did not produce any meaningful Ra values, as the trenched plots showed similar or even
higher $CO_2$ efflux rates than the untreated control plots.
Modelled Rh (Eq. 3) was slightly lower than field Rs during the cold season (Fig. 4). The gap between Rh and Rs
(measured and modelled) became larger during the growing season, implying highest contribution of Ra during the
warm monsoon months at both sites (Fig. 4 and 5). The modelling approach yielded generally higher annual
autotrophic contribution (Ra = 43 %, coniferous, 45 % broadleaved) when compared to the trenching experiment. At
the broadleaved forest, a larger fraction of Rh was attributed to the 0-10 cm mineral soil layer, whereas all three
layers (organic, 0-10, and 10-30) contributed similarly to Rh at the coniferous forest (Fig. 5). Modelled cumulative
annual (2015) Rh and Ra were 7.3 and 5.5 t C ha$^{-1}$ at the coniferous and 8.0 and 6.5 t C ha$^{-1}$ at the broadleaved forest
respectively.





**5. Discussion**
Our hypothesis, that Rs was higher at the lower elevation broadleaved forest site was not confirmed although soils
had been consistently about 4°C warmer than at the higher elevation coniferous forest. Annual Rs was similar among
both forest types (12.8 – 14.5 t C ha$^{-1}$) and was in the range of values reported for similar ecosystems (10.1-13 t C
ha$^{-1}$ (Dar et al., 2015); 10-12 t C ha$^{-1}$ (Li et al., 2008); 13.7 t C ha$^{-1}$ (Yang et al., 2007) and 14.7 t C ha$^{-1}$ (Wang et al.,
2010)). The higher altitude coniferous forest had double tree basal area and standing stock, indicating that this specific
forest type is exceptionally productive (Singh et al., 1994; Wangda and Ohsawa, 2006b; Sharma et al., 2010a; Tashi
et al., 2016). Soil C stocks of ~ 127 t ha$^{-1}$ (0-30 cm depth mineral soil) indicate that these mixed coniferous forests
are likely among those ecosystems with the highest C storage capacity in the eastern Himalayas (Wangda and
Ohsawa, 2006a; Sheikh et al., 2009; Dorji et al., 2014a; Tashi et al., 2016). High soil C contents and stocks were
reflected in generally higher basal respiration ($R_{10}$) at the coniferous forest explaining the comparatively high annual
Rs rates at this cooler, higher altitude, site.
At both forests, Rs followed the seasonal course of soil temperature and showed high apparent temperature
sensitivities (field $Q_{10}$ between 4 and 5). Field $Q_{10}$ values, however, not only reflect the effects of soil temperature
but manifest all interacting drivers of Rs throughout the season (Davidson and Janssens, 2006; Schindlbacher et al.,
2009; Ruehr and Buchmann, 2010). Soil temperature and soil moisture co-varied during both study years with dry
and cold winters and optimal soil moisture during the warm summer months. To account for this co-variation, we
normalized all Rs measurements using Eq. (2), to the corresponding optimal soil moisture of the sites (39 and 43 %)
and re-calculated the $Q_{10}$ values. Moisture normalization Rs had a $Q_{10}$ of ~ 3 at both sites which show that co-variation
between soil temperature and moisture was one reason for the high apparent temperature sensitivity of Rs. The
moisture normalized field $Q_{10}$ of ~ 3 came already closer to the intrinsic temperature sensitivity of Rh ($Q_{10}$ ~ 2.3 at
both sites) which was determined under controlled lab conditions at soil temperatures from 5 to 25 °C. Since $Q_{10}$ tend
to decrease with decreasing temperatures (Leifeld and Fuhrer, 2005; Tuomi et al., 2008; Schindlbacher et al., 2010),
we further calculated lab $Q_{10}$ at temperature ranges which came closest to the soil temperature range in the field (5-
15 °C coniferous, 5-20 °C broadleaved). As expected, $Q_{10}$ were slightly higher (2.6 ± 0.5 coniferous, 2.7 ± 0.3
broadleaved) as the ones calculated over the whole 5-25 °C range, but were still below the moisture normalized field
$Q_{10}$ values. The remaining difference between lab (Rh) and field (Rs) $Q_{10}$ can result from a higher apparent
temperature sensitivity of Ra (Boone et al., 1998) driven by accelerated below ground transport of labile C during
the growing season (Schindlbacher et al., 2009). Enhanced priming of SOM decomposition (Bader and Cheng, 2007;
Dijkstra and Cheng, 2007; Kuzyakov, 2010; Bengtson et al., 2012) during the growing season could further add to
the strong apparent temperature response of Rs.
$Q_{10}$ of Rh was similar between sites but decreased when the soil became dry. Such dry conditions were only reached
during winter, during which Rs was generally low. Our simple empirical temperature-driven Rs model explained
most of the temporal variation in Rs under typical monsoon weather patterns during 2014 and 2015. However,
monsoon failures and drought periods have occurred in the past and may even increase in frequency and/or severity
of climate change (Cook et al., 2010; Schewe and Levermann, 2012; Menon et al., 2013; Sharmila et al., 2015). To
model such drought effects, it is necessary to further develop the model by integrating potential soil moisture response





of Rs (as we already did for Rh). To do so, longer Rs time series which include dry years and/or data from artificial drought experiments are needed for model parameterization and testing. It is intended to continue Rs measurements at both sites and Rs data from the ongoing throughfall manipulation experiment is in preparation for further model development. Our last hypotheses that Rs decreased during water logging at peak monsoon was not confirmed as soils at both sites were well drained and water logging did not occur even during periods of high precipitation (Fig. 1).

The two approaches to estimate the autotrophic contribution to Rs performed differently. While the trenching method showed ambivalent outcome, the modeling approach did well. Modelled Rh in the field remained slightly below Rs during the cold season. This can be expected as the contribution of Ra is generally lower during the dormant season than during the growing season (Hanson et al., 2000; Rey et al., 2002)**.** During the growing season, the two-component model, Eq. (3) predicted an increase in the contribution of Ra. Such a pattern has frequently been observed in other forest ecosystems and reflects the higher downward allocation of labile C during the growing season. Our model estimated ~ 45 % contribution of Ra to Rs falls well within estimates from other forest sites. The modeling approach holds some uncertainty. C stocks from deeper soil layers were not accounted for and a single $Q_{10}$ (0-10 cm depth, lab incubation) was used for the whole 10 - 30 cm mineral soil layer. Furthermore, potential effects of priming were not accounted for in our modelling approach. In contrast to our modeling approach, which was based on incubation results and soil C stocks, trenching was applied as an attempt to estimate Ra *in situ*. The trenching method, although highly invasive, can provide reasonable estimates of Ra for several forest types (Hanson et al., 2000; Subke et al., 2006) considered that all the caveats of the method were accounted for. Our trenching approach, however, largely failed at both study sites. There might be several reasons, albeit the trenching effects on soil moisture, which we had accounted for. Fine roots can maintain respiration for a comparatively long time after cutting (Lee et al., 2003) and if roots die, their decomposition adds to the soil $CO_2$ efflux (Hanson et al., 2000). This was likely the main reason for the absence of any effect at plots which had been trenched during spring of the same year as of subsequent Rs measurements (year 2015). Plots which had been trenched one year earlier, already showed decreased Rs, but the estimated autotrophic contribution was on average < 30 % and highly variable. Considering a dead fine root mass loss of roughly one-third during the second year after trenching (Díaz-Pinés et al., 2010) and accounting for the corresponding effects on soil $CO_2$ efflux (additional efflux ~ 1 t C ha[-1]), the estimated contribution of Ra increased to ~ 40 % which is in the range of our modeling results.

Soil C input via aboveground litter-fall was almost similar between forest types (~ 3.5 t C ha[-1]) although tree basal area was substantially lower at the broadleaved forest. This can be attributed to a generally higher leaf litter production in broadleaved ecosystems (Bisht et al., 2014; Tiwari and Joshi, 2015). Fine root stocks at both forest types fall within the upper range of estimates from other surveys in the Himalayan region (Adhikari et al., 1995; Usman et al., 1999; Garkoti, 2008; Rana et al., 2015), especially if it is considered that fine root contents in this study were estimated solely for 0-30 cm mineral soil depth. Assuming a mean fine root turnover time of one year (Brunner et al., 2013), the annual fine root litter input from 0-30 cm soil layer was ~ 2 and ~ 3 t C ha[-1] at the coniferous and broadleaved forest, respectively. During 2015, the estimated soil C input (leaf litter and fine root litter of the top 30 cm soil) was, therefore, ~ 1.5 tons lower than the estimated annual gaseous soil C loss via Rh. This, however, is only



a first rough approximation of the real soil C budget, since fine root turnover was not adequately determined and
important C fluxes, such as for instance, DOC leaching, root litter production below 30 cm depth, and C input from
vigorously growing herbaceous ground vegetation were not accounted for in our study, which primarily aimed at a
detailed characterization of the soil $CO_2$ efflux.

**6. Conclusion**


The monsoon climate allows for highly productive mountain forests in the eastern Himalayas. Such forests can store
high amounts of C in plant biomass and soil, which was particularly evident in the high altitude coniferous forest in
our study. The high-temperature sensitivity of Rs ($Q_{10}$ 4-5) suggests that soil C cycling could react particularly
vulnerable to global warming. Deeper analyses, however, showed that Rh had similar temperature sensitivities as
other forest soils ($Q_{10}$ 2-3) and that co-variation of soil moisture and Ra led to the high field $Q_{10}$. At both forests
studied, a simple temperature-driven model was sufficient to explain most of the temporal variation in Rs during the
two study years. Both study years had typical monsoon climate with dry and cold winters and monsoon rain during
the warm season. Further research and model development is, however, warranted to better understand how
infrequent/extreme events such as monsoon failure and drought affect soil/ecosystem C cycling and Rs in these forest
ecosystems.

**7. Author contribution**


N. Wangdi carried out the research and data analysis and drafted the manuscript. M P. Nirola carried out the
incubation experiment and analysed the data. N. Zangmo and K. Orong collected the data and continuously monitored
the research sites. I.U Ahmed carried out the root and the soil sampling study within our research sites. M. Mayer
performed modelling and contributed to writing the manuscript. A. Darabant, R. Jandl, G. Gratzer designed the
experiment and provided feedback on the manuscript. A. Schindlbacher supervised the overall work, designed the
experiment and critically revised the manuscript.

**8. Acknowledgements**


We are highly grateful to the all the staff of Renewable Natural Resources Research and Development Centre,
Yusipang for all the support for the field work. We also thank the management and staff of the Ugyen Wangchuck
Institute for Conservation and Environment, Bumthang for providing additional support for the study. This study was
part of the work package I of the BC-CAP project (Climate Change Adaptation potentials of forests in Bhutan –
Building human capacities and knowledge base) jointly implemented by the Department of Forest and Park Services,
Bhutan and University of Natural Resources and Applied Life Sciences (BOKU), Austria with funding from
Government of Austria, through the Ministry of Agriculture, Forestry, Environment and Water Management.




**9. Disclaimer**

The views and opinions expressed in this article are those of the authors and do not necessarily reflect the views of

any institutions of the Royal Government of Bhutan or the Government of Austria.





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





**Table 1 Site characteristics of the two studied forests types**

| Parameter | Coniferous forest | Broadleaved forest |
|---|---|---|
| Elevation (m) | 3260 | 2460 |
| Latitude | 27°28' 00" N | 28°28'51.06" N |
| Longitude | 89°44'30.79''E | 89°51'27.73" E |
| Annual Precipitation 2015 (mm) | 1167 | 1120 |
| Mean Air Temperature 2015 (∘C) | 7.8 | 12.0 |
| Dominant Overstorey species | | *Quercus lanata (63.5%)* |
| | *Tsuga dumosa (59.5%)* | *Quercus griffithii (29.6%)* |
| | *Quercus semecarpifolia (29.3%)* | |
| | *Picea spinulosa (6.3%)* | |
| | *Abies densa (4.1%)* | |
| | *Taxus baccata (0.3%)* | |
| Dominant Understory species | *Ilex dipreyana (0.2%)* | *Symplocus sp. (0.8%)* |
| | *Rhododendron arboreum (0.1%)* | *Lyonia ovalifolia, (2.2%)* |
| | | *Rhododendron arboreum (3.4%)* |
| Tree density (No. ha$^{-1}$) | 364 ± 50 | 569 ±19 |
| Mean Tree height (m) Overstorey | 24.4 ± 2.1 | 23.6 ± 1.4 |
| Mean Tree Height (m) Understorey | 7.8 ± 3.5 | 9.8 ± 0.4 |
| Mean DBH (cm) Overstorey | 50.7 ± 5.8 | 37.8 ±2.3 |
| Mean DBH (cm) Understorey | 13.8 ± 1.4 | 16.1 ± 0.9 |
| Tree basal area (m$^2$ ha$^{-1}$) | 77.5 ± 4.6 | 39.9 ± 4.4 |
| Standing volume (m$^3$ ha$^{-1}$ ) | 1066 ± 2.3 | 464 ± 25 |
| Soil organic carbon (t ha$^{-1}$) 0-30 cm | 127.2 ± 17.4 | 91.2 ± 6.2 |
| Soil organic nitrogen (t ha$^{-1}$) 0-30 cm | 6.8 ± 0.6 | 4.2 ± 0.1 |
| pH   (0-10 cm) | 5.2 ± 0.1 | 5.0 ± 0.1 |
| Bulk density (g cm$^{-3}$) 0-10 cm | 0.61± 0.02 | 0.61 ± 0.01 |
| Fine Root biomass (t C ha$^{-1}$) 0-30 cm | 2.3  ± 0.3 | 3.2 ± 0.5 |
| Litter input (t C ha$^{-1}$ yr $^{-1}$) | 3.5 ± 0.10 | 3.4 ± 0.03 |

*All stand and soil parameters are expressed as the mean ± standard error.












| Site | Soil moisture levels | Temperature Range (°C) | $Q_{10}$ | $R_{10}$ (µmol $CO_2$ $kgC^{-1}$ $S^{-1}$) |
|---|---|---|---|---|
| **Coniferous forest** | Intermediate (32 Vol. %) | 5.0 - 25.0 | $2.58 \pm 0.22$[a] | $0.09 \pm 0.01$[a] |
| | Dry (12 Vol. %) | 5.0 - 25.0 | $1.59 \pm 0.07$[b] | $0.07 \pm 0.00$[a] |
| | Wet (56 Vol. %) | 5.0 - 25.0 | $2.12 \pm 0.14$[a] | $0.12 \pm 0.01$[b] |
| **Broadleaved forest** | Intermediate (31 Vol. %) | 5.0 - 25.0 | $2.35 \pm 0.09$[a] | $0.13 \pm 0.01$[a] |
| | Dry (12 Vol. %) | 5.0 - 25.0 | $1.60 \pm 0.11$[b] | $0.09 \pm 0.01$[a] |
| | Wet (55 Vol. %) | 5.0 - 25.0 | $2.05 \pm 0.14$[a] | $0.16 \pm 0.01$[b] |

**Table 2. $Q_{10}$ and $R_{10}$ parameters (Eq. (1)) derived from laboratory incubation. Letters indicate significant**
**differences in $R_{10}$ and $Q_{10}$ between soil moisture levels of the mineral soil samples.**



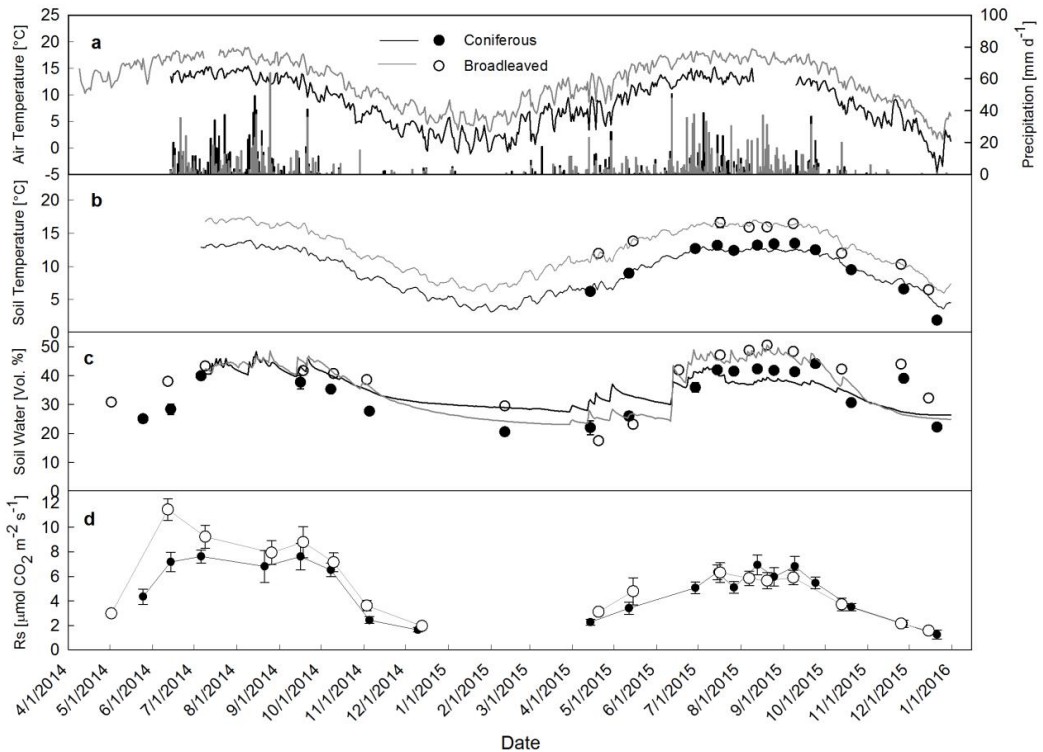


**Figure 1. Seasonal course of air temperature and precipitation (a), soil temperature (b), volumetric soil water content (c), and soil respiration (d) measured at a coniferous and a broadleaved forest in Bhutan Himalayas in 2014 and 2015. Circles represent daily mean values of manual measurements. Solid lines (a, b, c) represent daily mean values of continuous measurements. Error bars indicate standard error of the mean.**





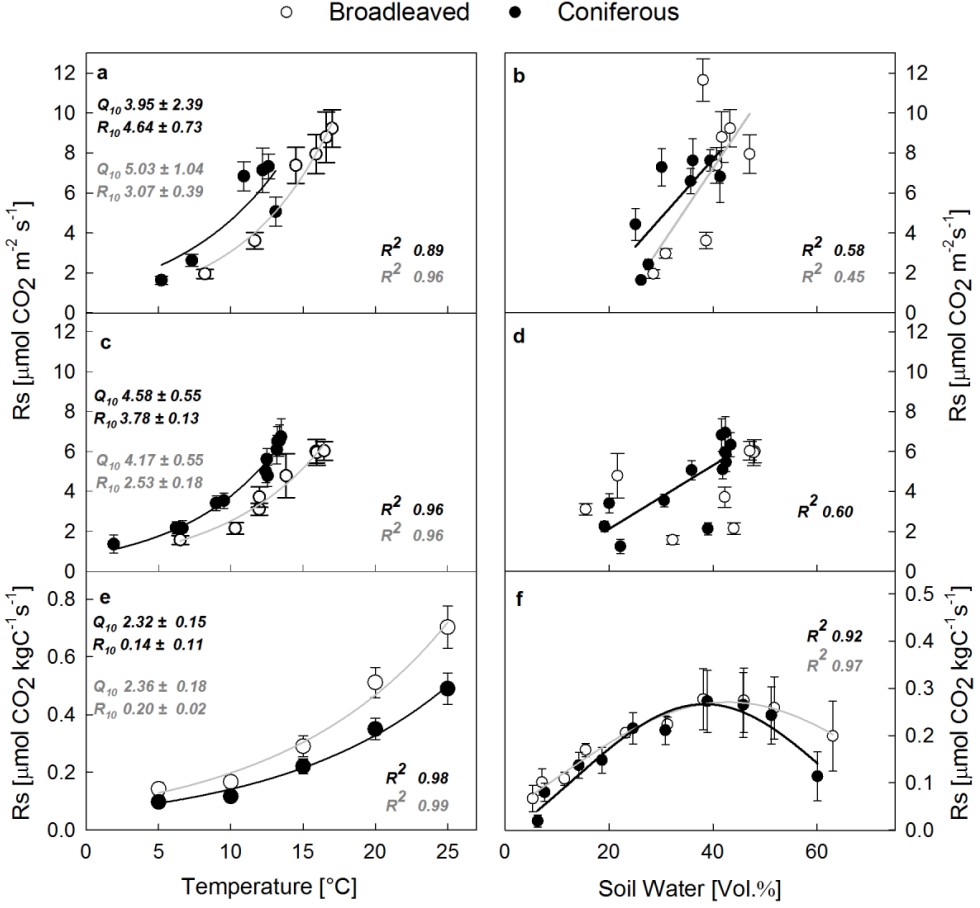

**Figure 2.** Relationship between soil $CO_2$ efflux and soil temperature and soil water content for a broadleaved forest and a coniferous forest in Bhutan Himalayas, measured during field campaigns in 2014 (a, b) and 2015 (c, d), and during a laboratory incubation experiment (e, f). Relations between $CO_2$ efflux and temperature were fitted with an exponential function, Eq.(1) and model parameters ($R_{10}$, $Q_{10}$) are shown. Relations between $CO_2$ efflux and volumetric water content were fitted with linear and Gaussian functions Eq. (2). Error bars represent standard error of the mean.





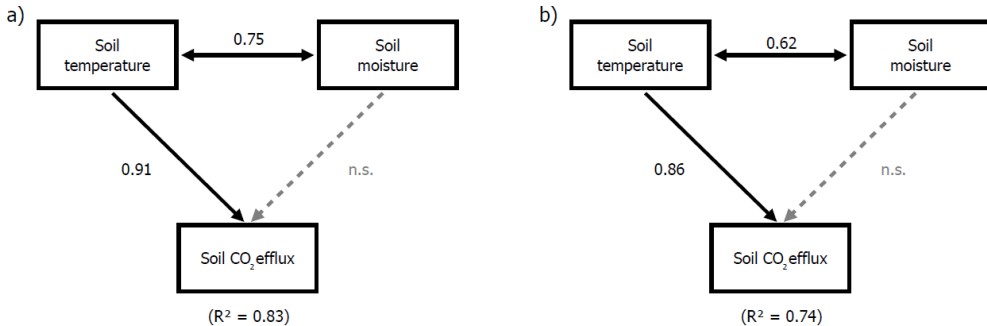


**Figure 3. Structural equation models for a broadleaved (a) and a coniferous (b) forest in Bhutan Himalayas, describing the soil climatic drivers of soil CO₂ efflux during measurement campaigns in 2014 and 2015.**



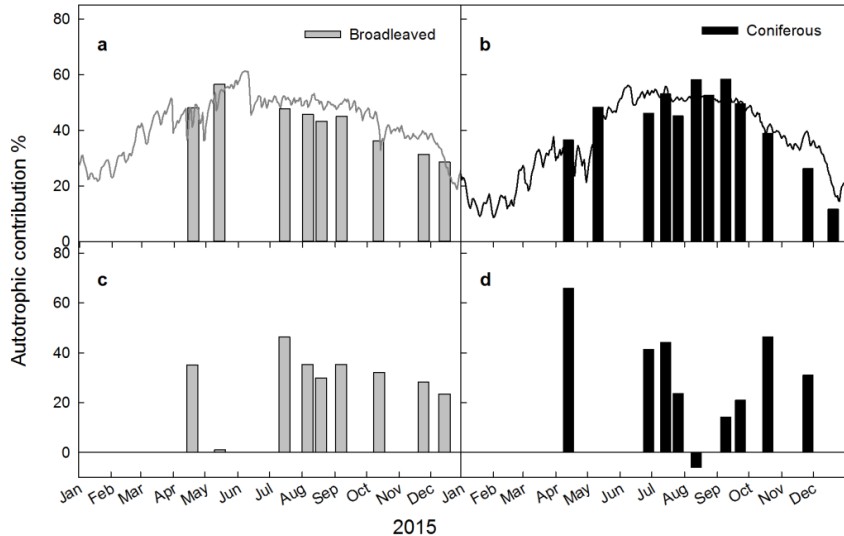


**Figure 4: Monthly contribution of autotrophic soil respiration to total soil $CO_2$ efflux at a broadleaved and coniferous**
**forest in Bhutan Himalayas. Data on autotrophic respiration are derived from the difference of modelled daily soil $CO_2$**
**efflux and modelled heterotrophic soil respiration rates (a, b, solid lines), measured soil $CO_2$ efflux and modelled**
**heterotrophic soil respiration rates (a, b, bars), and measured soil $CO_2$ efflux from control and trenched plots (c, d) of the**
**respective site.**





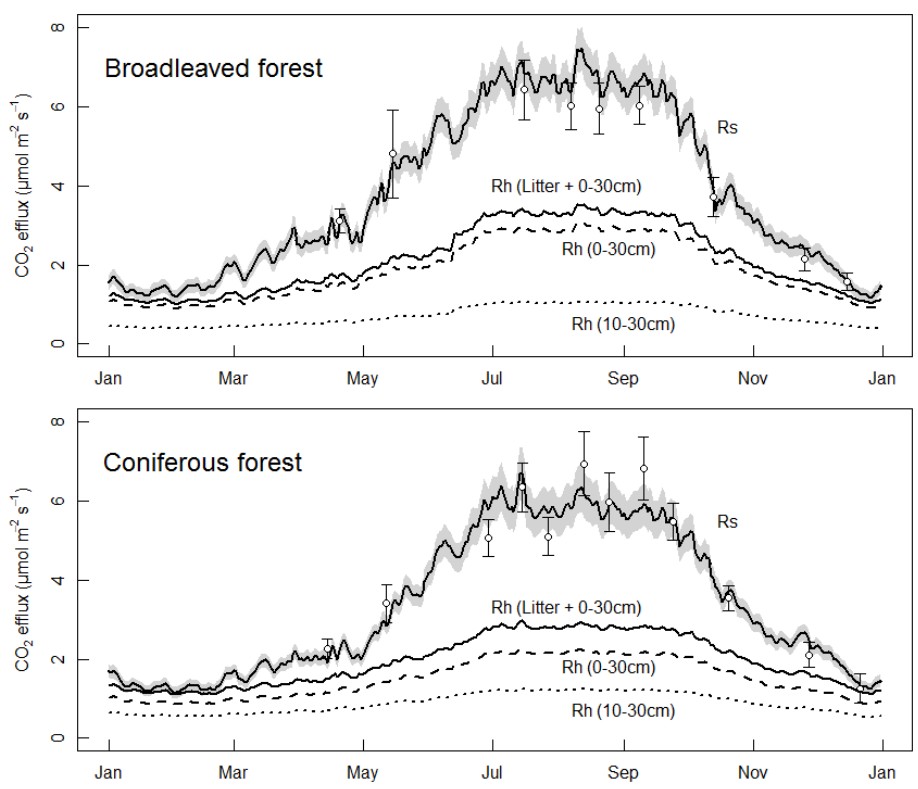


**Figure 5: Seasonal course of modelled soil $CO_2$ efflux (Rs) and heterotrophic soil respiration rates (Rh) from different soil**

**layers at a broadleaved and coniferous forest in Bhutan Himalayas in 2015. Open circles are measured soil $CO_2$ efflux**
**rates. Error bars and shaded area represent standard error of the daily mean.**