# Peer review of "Figure S1: Response curves of relative CO2 efflux (heterotrophic soil respiration) to volumetric soil water content derived from a laboratory incubation (see Materials and methods section for details)."

_Biogeosciences, 2016_

## Referee Comment (RC1) · Anonymous Referee #1 · 16 Aug 2016

General comments
* * *
This manuscript describes measurements of soil respiration (Rs) made in the field in two forests in Bhutan, supplemented by laboratory incubations, that looks at Rs fluxes, sources, seasonally-driven changes, as well as sensitivities to temperature and moisture. This is interesting and valuable, given the paucity of data from this region, if not groundbreaking. The ms is reasonably well written, although there are many minor English errors, and frequently insightful. I particularly liked the comparison between different techniques and measurements/models, even though this is not developed as fully as it could be.

There are some problems. Aspects of the methods and results are unclear; in particular, the authors should be careful to distinguish between measured and modeled results, why and when each was performed, and when they're referring to each. I think the trenched plot results could be better discussed, particularly as the problems that occurred seem relatively straightforward to explain. Finally, and very importantly, the authors need in my opinion to post or include the data and code backing all their main results (see #9 below).

In summary, this ms needs moderate to significant revisions in many places, but it's fundamentally a strong and interesting study.

Specific comments
* * *
1. Lines 23-24: why does a discrepancy between modeled and measured Ra indicate trenching performed poorly? Clarify. Might also add ", probably because of the short time lag between trenching and measurement"?

2. L. 27: "preceding" ?

3. L. 37: maybe "potentially feeds back on global climate change". Also there are better citations for this, e.g. Frey et al. 2013 (10.1038/nclimate1796) or Wang et al. 2014 (10.1111/gcb.12620)

4. L. 50: start a new paragraph

5. L. 67: "would show decreases in Rs during"

6. L. 118-: what was delay between trenching and starting measurements?

7. L. 172: better to say "Effects of site" rather than "forest type" since you can't actually test forest type (as n=1)

8. L. 190: perhaps "to calculate a projected daily field Rs" for clarity

9. Availability of code and data? It's 2016, and I expect all code and data (at least that backing the main results) to be included as supplementary info, or posted in a repository. It's not acceptable to produce results from a black box 10. L. 252-257: interesting!

11. L. 278: "any meaningful Rh values"?

12. L. 310-: "Q10 tends"; also should be "increase with decreasing temperatures"?

13. L. 314: "than the ones"

14. L. 327: "We intend to"

15. L. 333: this is awkward English and unclear – why "ambivalent"?

16. L. 338: "Rs, falling well within"

17. L. 345: "albeit"? What?

18. L. 348-353: interesting though unsurprising. Might mention this in abstract

19. Figure 1: minor point but perhaps format x axis dates as "Apr 2014", "May 2014", or something like that to eliminate M/D/Y ambiguity (i.e. make consistent with Figures 4 and 5)

20. Figure 3: necessary?

21. Figure 4: this is confusing. At the very least, clarify the caption, and perhaps re-think how these data are displayed

22. Figure 5: perhaps note in caption that the Rh lines are cumulative

---

## Referee Comment (RC2) · Anonymous Referee #2 · 10 Oct 2016

The paper "Soil CO2 efflux from two mountain forests in the Eastern Himalayas Bhutan: components and controls" by Wangdi et al. provides further information on a data poor environment, relevant to the Earths carbon budget. In this way the paper is a useful contribution to the cannon. Further lab-based incubations also appear useful for constraining modeled behaviors in the field, and the provided comparisons to in situ outcomes may be informative.

I have a number of concerns about the choice to include some of the provided data, as well as the exact equations and parameters that the model utilized to determine partitioning between respiration components. Overall, in agreement with the first reviewer, I believe that this study has some sound and useful information and analysis that should

be published, but I would expect major revisions would be necessary to clear out the unnecessary components of the article and clarify others.

My broad concerns will be listed first with specific items listed afterwards:

1) The "coniferous forest" as described has a substantial component of broadleaved trees (∼29% Quercus sp.), and is later described as, perhaps more appropriately, a "cool temperate mixed coniferous forest" (line 72/73). Perhaps describing as a "mixed forest" throughout the paper would be more helpful. This is likely to have some impact on respiratory fluxes (through litter quality, leaf economics, etc) and this, along with the potential impacts from soil type and understory plant types, density and behavior is not addressed in the text sufficiently in my mind.

2) I question the value of the 2014 field-based Rs results, considering that they are acknowledged by the authors to be influenced by pressure effects from chamber placement.

3) I am uncomfortable with different mathematical functions being used to determine the same biological functionality (in particular the linear versus Gaussian response of soil water content to respiration rates). I would prefer that whichever function is used that there is some biological rationale that can be used to defend this choice.

4) I agree with the first reviewer that it would be better to have the model by which Rh and Ra components were calculated either explained through the primary equations in the text, or by incorporating the model as a supplementary material.

5) The use of the term Q10 to describe the entire soil response to, effectively, seasonal changes is inappropriate to my mind. By definition Q10 refers to the change in reaction rate of an enzyme or system to 10 degree changes in temperature, and on this basis the lab-based incubation Q10s are appropriate and should be retained and used in the models, but calling the whole system response a Q10 when the authors acknowledge (lines 301-303) that it incorporates water content, leaf litter availability and other covariable parameters makes this use of the Q10 term meaningless.

—————————————————————— 1) Line 25/26: see broader point 5 above. This is not in any way a Q10 with the number of conflating variables. Please use different terminology.

2) Lines 64-67: These hypotheses are not all that useful and the final hypothesis is not addressed within the paper, leading to a question of whether these are needed in the paper at all.

3) Line 78: Acer campbelli is listed as a dominant species in the cool, temperate mixed coniferous forest but is not listed in Table 1.

4) Lines 88-92: Climate can vary dramatically in mountainous regions over spatial scales of 1km. Is there evidence that these weather stations were recording appropriate data for these sites?

5) Line 123: By the nature of its close follow on after trenching this seems to refer to volumetric soil water measurements in the trenched plots but instead refers to the broader study plots (as shown in Figure 1). This could be more clear.

6) Lines 145-146: I would be interested in hearing more about the ventilation system used for the incubations. I am uncertain how much water might be lost by the soils during this process (e.g.- the ventilation process during the two-week waiting periods between soil moisture sampling) and how this water loss was addressed during periods between measurements.

7) Lines 143-148: I wonder about the effect of sieving on Rh considering the disruption placed on the soil/fungal community. It seems likely that this has significantly affected this component within this aspect of the study. (Datta et al Int. Agrophys., 2014, 28, 119-124)

8) Agreed with reviewer #1 point 7

9) Lines 188-194: The assumption that the temperature in the soil at 5cm depth is sufficiently predictive of Rs may work within this model but it assumes that the system is sufficiently co-variant that this one data point is essentially all that is needed. This seems to assume that the basal respiration from lower soil depths is effectively constant. Can the authors provide any evidence that this is true?

10) Line 200: I agree that a Gaussian distribution is probably the most appropriate here (and for appropriate biologically relevant reasons) but the linear fits later in figure 2 have no real biological rationale.

11) Lines 205-212: The trenching experiment not only affects water retention in the soil but also provides further litter availability and there are likely non-linear effects that are not well addressed in this section. I am also unconvinced that the correction for soil moisture is precise and accurate based upon the data reported. Perhaps it would be more useful to report a range of possible outcomes instead of the firm values reported here.

12) Line 220: It is unclear if the lack of specific moisture response function is due to a lack of (or no) collected data or a poor linear or Gaussian fit was obtained from the collected data.

13) Line 246-247: The method for assessing fine root biomass is not reported. Either the method should be discussed or a reference to the data would be helpful.

14) Lines 252-254: Given the potentially compromised nature of the Rs data from 2014 I would prefer that it not be reported at all, especially given the successful campaign run through 2015. The nature of pressure pumping and its effects on fluxes is sufficiently well established that this doesn't add much value to the paper.

15) Line 259: This is somewhat self-fulfilling. You measured once every three weeks and find that 3 week sampling density is sufficient. In order to truly test this you would need a higher density sampling rate that you are then able to sub-sample at the 3 week

frequency. I would suggest this comment (and others similar) be removed from the text.

16) Line 277-278: Again, given the nature of the trenched plots in 2015 (errors in strategy that are explainable and understandable) I am uncertain why this is discussed in the methods section and here. If I understand correctly not including this would save space and would not affect your analysis.

17) Lines 280-287: The model should be made more clear, in agreement with point 9 from reviewer #1.

18) Lines 301-319: I find this justification of the "field Q10" values to be unconvincing and suggest that this section be reworked or removed from the text. There are too many other variables that are not addressed beyond the already tenuous soil moisture correction for this to be adequately compared to a true Q10.

I agree that Figure 3 seems to serve little purpose and any lost detail can be described quickly and easily in the text.

---

## Referee Comment (RC3) · Anonymous Referee #3 · 11 Oct 2016

This manuscript entitled "Soil CO2 efflux from two mountain forests in the Eastern Himalayas Bhutan: components and controls" by Wangdi et al. provides interesting, relevant and valuable information on a poorly studied region. Manuscript is mostly well written and easy to read. The use and comparison of different techniques of measurements and different models is interesting. Nevertheless, aspects of the methods and then of the results remained unclear because it was not easy to distinguish and understand when and also why measured or modeled results were used to suit the purpose.

Major revisions would be necessary to clarify the manuscript and to develop more explicitly the objectives of the comparison between different measurements/models.
General comments ———————————- I am not convinced that 2014 field Rs data should be presented in the ms as they are not relevant because influenced by pressure effects. In the same way given the trenched plots in 2015 didn't produced meaningfull values, what does these data bring to the analysis? If retained, the trenched plot results could be better discussed.

Important care must be given to distinguish between measured and modeled results. Authors should explain why and when each was performed and also why and when they are referring to each

Specific comments ———————————- 1) l.23-24 : unclear. I can't see why the variability of Ra indicates a methodological issue with the trenching

2) l.272 : prefer effect of sites rather than of forest type

3) l.190 : discuss how constraining the model with the temperature in the soil at 5cm depth is sufficient and relevant. What about the deepest contributions to Rs ?

4)l.190 : The same parameters (of Eq1) are used to model Rs over the year without any discussion whether or not the Q10 could vary with the temperature range over the year.

5)l.205-212 : agreed with reviewer #2 point 11. Indicate the uncertainties rather than that corrected value.

6) l.218: what is Fig S1 ?

7) l.246 : report and discuss the method used to estimate fine root biomass

8) l.259 : How can you be convinced that it 'indicates that a three-week interval is sufficient' although you didn't measured with a higher frequency ? Restrain the purpose.

9) l.278 : useful ?

10) l.308-319: I have issues with the analysis presented here because I am concerned

about the definition for the terms intrinsic and apparent sensitivities. Recently, Sierra et al. 2015, JAMES 7: 335-356 proposed consistent and formal definitions for intrinsic and apparent sensitivity. It would be nice if the authors referred to that definition or explained how they defined these conceptual sensitivities.

11) l.345: albeit ?

12) Figure 4: The figure is really confusing. Caption doesn't help. . .

13) Figure 5: not easy to understand that the lines are cumulative. Indicate by filling with different colors that the bottom area is Rh (10 – 30), the second area is Rh (0 – 10), the third (upper) one Rh litter and the highest Ra.
* * *

---

## Short Comment (SC1) · 18 Oct 2016

General response to reviewer comments:

We highly appreciate the effort and the constructive suggestions of our reviewers and the section editor!

All three reports point at the need for a better description of the Rh model, for better distinguishing between model results and measured data in the text, for a better definition and consistent use of terms with regard to temperature sensitivity, and for a better explanation/discussion of the trenching results. We fully agree with our referees in most points and we will adapt the manuscript accordingly. We will add the model-

[Figure]

R-script and the measured data as supplements. Regarding sensitivity, we will stick to the definitions by Davidson and Janssens 2006 and Sierra et al. 2015, respectively. We agree that field Rs is not driven by temperature alone and therefore, we will avoid the term "Q10" for the relationship between field Rs and field soil temperature. We will remove all 2014 Rs data, as suggested (pressure problem during measurements) and improve Fig. 4 (trenching), which indeed is difficult to understand. We will remove Fig 3. This information can be brought in the text. We will add more discussion with regard to issues associated with trenching. Reviewers 2 and 3 suggested removing all information about the added (2015) trenching plots which produced unusable data. We would prefer keeping the information in the paper. It could be helpful for readers to be pointed to potential shortcomings/pitfalls of this method. For us, the persistently high CO2 fluxes after trenching came somewhat unexpected. Therefore, we would prefer sharing these observations.

We will revise the manuscript within the coming weeks and send a detailed response letter to each reviewer.

---

## Author Comment (AC8) · 25 Nov 2016

Anonymous Referee #3 This manuscript entitled "Soil CO2 efflux from two mountain forests in the Eastern Himalayas Bhutan: components and controls" by Wangdi et al. provides interesting, relevant and valuable information on a poorly studied region. Manuscript is mostly well written and easy to read. The use and comparison of different techniques of measurements and different models is interesting. Nevertheless, aspects of the methods and then of the results remained unclear because it was not easy to distinguish and understand when and also why measured or modeled results were used to suit the purpose.

Major revisions would be necessary to clarify the manuscript and to develop more

explicitly the objectives of the comparison between different measurements/models.

We highly appreciate the constructive comments which largely aligned with suggestions from reviewers 1 and 2. We better distinguish between modeled Rh and trenching results in the revised manuscript and we better explained why these methods were used. All other suggestions have been followed as well. While revising the manuscript, we found a minor conversion error for Rh values per kgC-1, which we corrected for in the revised manuscript. Corrections did not affect the overall outcome of the study. Corrections resulted in slightly higher modeled Rh and minimal, insignificant deviations of Q10 and R10 values when compared to the values in the initial manuscript (without any effect on the study results!).

General comments:

I am not convinced that 2014 field Rs data should be presented in the ms as they are not relevant because influenced by pressure effects. In the same way given the trenched plots in 2015 didn't produced meaningful values, what does these data bring to the analysis? If retained, the trenched plot results could be better discussed. Important care must be given to distinguish between measured and modeled results. Authors should explain why and when each was performed and also why and when they are referring to each.

As suggested, we removed the 2014 Rs data as well as the data from 2015 trenching plots. The text is now easier to read and we did not lose relevant information.

Specific comments:

1) l.23-24 : unclear. I can't see why the variability of Ra indicates a methodological issue with the trenching

We removed this from the abstract. We further refined the discussion and point at shortcomings of both methods (model, trenching) in the discussion section (L259-303).

2) l.272 : prefer effect of sites rather than of forest type

Changed to "sites" in the whole manuscript

3) l.190 : discuss how constraining the model with the temperature in the soil at 5cm depth is sufficient and relevant. What about the deepest contributions to Rs?

We actually used soil temperatures from different soil layers (mineral soil 5 cm, and mineral soil 20 cm depth) for modeling (L180-183) of $CO_2$ efflux from the corresponding layers. $CO_2$ efflux from < 30 cm depth was neglected in our model. We discussed this in terms of the Rh model outcome. We extended the discussion accordingly (L259-284).

4)l.190 : The same parameters (of Eq1) are used to model Rs over the year without any discussion whether or not the Q10 could vary with the temperature range over the year.

We alternatively fitted a Gaussian function (where Q10 changes with temperature). The fit of the simple exponential function was slightly better. We therefore decided to stick to this function.

5)l.205-212 : agreed with reviewer #2 point 11. Indicate the uncertainties rather than that corrected value.

As both methods have some different sources of uncertainty, they are quite difficult to quantify. We therefore stuck to the graph, but better discuss uncertainties in the text.

6) l.218: what is Fig S1 ?

Supporting Figure 1 is a supplement. We deleted this figure as this is a common procedure in model anyway.

7) l.246 : report and discuss the method used to estimate fine root biomass

We added the method (L66-70).

8) l.259 : How can you be convinced that it 'indicates that a three-week interval is sufficient' although you didn't measured with a higher frequency ? Restrain the purpose.

That's true. We deleted this sentence.

9) l.278 : useful ?

Deleted.

10) l.308-319: I have issues with the analysis presented here because I am concerned about the definition for the terms intrinsic and apparent sensitivities. Recently, Sierra et al. 2015, JAMES 7: 335-356 proposed consistent and formal definitions for intrinsic and apparent sensitivity. It would be nice if the authors referred to that definition or explained how they defined these conceptual sensitivities.

We completely revised the section. According to reviewer 2, we don't use Q10 for field Rs. The text passage is shorter and much easier to read now. We refer to Sierra et.al. in the discussion section with regard to moisture effects on temp sensitivity.

11) l.345: albeit ?

Should be "besides" – changed.

12) Figure 4: The figure is really confusing. Caption doesn't help

We adapted the figure. Should be easy to understand now.

13) Figure 5: not easy to understand that the lines are cumulative. Indicate by filling with different colors that the bottom area is Rh (10 – 30), the second area is Rh (0 –10), the third (upper) one Rh litter and the highest Ra.

We adapted the figure and caption. It should be clear now.

Please also note the supplement to this comment:
http://www.biogeosciences-discuss.net/bg-2016-291/bg-2016-291-AC8-supplement.pdf

[Figure]

**Supplement:**

##########################################################################

**Soil CO2 efflux from two mountain forests in the Eastern Himalayas Bhutan:**
**components and controls**

**by Wangdi et al.**

**R code for modelling heterotrophic respiration by means of laboratory**
**incubation data, soil carbon (C) stocks and continuous soil climate data**

##########################################################################

**Set working directory**

setwd("D:/Buthan/FinalVersion")

**Read continous soil climate data**

**Broadleaved forest**

BF = read.csv("SoilClimate_BroadleavedForest.csv", sep = ";")

**Mixed forest**

MF = read.csv("SoilClimate_MixedForest.csv", sep = ";")

**Variable description:**

**site: BF = broadleaved forest; MF = mixed forest**
**VWC_5cm: volumetric soil water content (vol.%) measured in 5 cm depth**
**VWC_20cm: volumetric soil water content (vol.%) measured in 20 cm depth**
**T_5cm: soil temperature (°C) measured in 5 cm depth**
**T_20cm: soil temperature (°C) measured in 20 cm depth**

```
**All values are daily mean values**

**Set parameters**

**Temperature response function (Equation 1)**

**Broadleaved forest**

**Forest floor litter**

BF_Lit_T_b0 = 0.265
BF_Lit_T_b1 = 0.0793

**Mineral soil**

BF_Min_T_b0 = 0.0961
BF_Min_T_b1 = 0.0828

**Mixed forest**

**Forest floor litter**

MF_Lit_T_b0 = 0.548
MF_Lit_T_b1 = 0.0645

**Mineral soil**

MF_Min_T_b0 = 0.0701
MF_Min_T_b1 = 0.0808

**Moisture response function (Equation 3)**
```

**Broadleaved forest**

**Mineral soil only**

BF_M_b0 =  0.0080456

BF_M_b1 =  0.01194

BF_M_b2 = -0.00012588

**Mixed forest**

**Mineral soil only**

MF_M_b0 = -0.086751

MF_M_b1 =  0.017487

MF_M_b2 = -0.00020757

**Soil moisture (vol.%)of samples during first incubations (see Table 2)**

**Broadleaved forest**

**Forest floor litter**

BF_M_Inc_Lit = 46

**Mineral soil**

BF_M_Inc_Min = 35

**Mixed forest**

**Forest floor litter**

MF_M_Inc_Lit = 46

**Mineral soil**

MF_M_Inc_Min = 33

**Soil carbon stocks (kg/m²) (see Table 1)**

**Broadleaved forest**

**Annual litter input (proxy for litter C stock)**

BF_C_Lit = 0.34

**Mineral soil C stocks in 0-10 cm depth (data from 4 soil pits)**

BF_C_Min_0_10 = c(6.04, 7.00, 5.46, 3.71)

**Mineral soil C stocks in 10-20 cm depth (data from 4 soil pits)**

BF_C_Min_10_30 = c(3.69, 4.01, 3.04, 3.10)

**Mixed forest**

**Annual litter input (proxy for litter C stock)**

MF_C_Lit =  0.35

**Mineral soil C stocks in 0-10 cm depth (data from 4 soil pits)**

```r
MF_C_Min_0_10 = c(5.34, 6.71, 4.83, 7.28)

**Mineral soil C stocks in 10-20 cm depth (data from 4 soil pits)**

MF_C_Min_10_30 = c(8.53, 7.46, 6.16, 9.91)

**Modelling heterotrophic respiration (Rh) for each layer**

**Broadleaved forest**

BF_matrix = matrix(nrow=nrow(BF), ncol=4)  # empty matrix

BF_matrix_mineral = matrix(nrow=nrow(BF), ncol=4)

BF_matrix_mineral_10_30 = matrix(nrow=nrow(BF), ncol=4)

for (i in 1:nrow(BF)) {

  for (j in 1:4) {

    BF_matrix[i,j] =

      # Forest floor litter

      (((BF_Lit_T_b0 * exp(BF_Lit_T_b1 * BF$T_5cm[i])) * # Modell Rh with Temp

      BF_C_Lit * # Multiply by C stocks

      ((BF_M_b0 + BF_M_b1 * BF$VWC_5cm[i] + BF_M_b2 * BF$VWC_5cm[i] ^ 2) / # Correct for
field moisture
```

(BF_M_b0 + BF_M_b1 * BF_M_Inc_Lit + BF_M_b2 * BF_M_Inc_Lit ^ 2)))

+

**Mineral soil 0-10 cm depth**

((BF_Min_T_b0 * exp(BF_Min_T_b1 * BF$T_5cm[i])) *

BF_C_Min_0_10[j] *

((BF_M_b0 + BF_M_b1 * BF$VWC_5cm[i] + BF_M_b2 * BF$VWC_5cm[i] ^ 2) /

(BF_M_b0 + BF_M_b1 * BF_M_Inc_Min + BF_M_b2 * BF_M_Inc_Min ^ 2)))

+

**Mineral soil 10-30 cm depth**

((BF_Min_T_b0 * exp(BF_Min_T_b1 * BF$T_20cm[i])) *

BF_C_Min_10_30[j] *

((BF_M_b0 + BF_M_b1 * BF$VWC_20cm[i] + BF_M_b2 * BF$VWC_20cm[i]^2) /

(BF_M_b0 + BF_M_b1 * BF_M_Inc_Min + BF_M_b2 * BF_M_Inc_Min ^ 2))))

BF_matrix_mineral[i,j] =

**Mineral soil 0-10 cm depth**

(((BF_Min_T_b0 * exp(BF_Min_T_b1 * BF$T_5cm[i])) *

BF_C_Min_0_10[j] *

((BF_M_b0 + BF_M_b1 * BF$VWC_5cm[i] + BF_M_b2 * BF$VWC_5cm[i] ^ 2) /

(BF_M_b0 + BF_M_b1 * BF_M_Inc_Min + BF_M_b2 * BF_M_Inc_Min ^ 2)))

+

**Mineral soil 10-30 cm depth**

((BF_Min_T_b0 * exp(BF_Min_T_b1 * BF$T_20cm[i])) *

BF_C_Min_10_30[j] *

((BF_M_b0 + BF_M_b1 * BF$VWC_20cm[i] + BF_M_b2 * BF$VWC_20cm[i]^2) /

(BF_M_b0 + BF_M_b1 * BF_M_Inc_Min + BF_M_b2 * BF_M_Inc_Min ^ 2))))

BF_matrix_mineral_10_30[i,j] =

**Mineral soil 10-30 cm depth**

((BF_Min_T_b0 * exp(BF_Min_T_b1 * BF$T_20cm[i])) *

```r
      BF_C_Min_10_30[j] *

      ((BF_M_b0 + BF_M_b1 * BF$VWC_20cm[i] + BF_M_b2 * BF$VWC_20cm[i]^2) /

      (BF_M_b0 + BF_M_b1 * BF_M_Inc_Min + BF_M_b2 * BF_M_Inc_Min ^ 2)))

    }
     }

**Calculate mean values in µmol CO2 kgC^-1 sec^-1**

BF$Rh = apply(BF_matrix, 1, FUN = mean)

BF$Rh_Min = apply(BF_matrix_mineral, 1, FUN = mean)

BF$Rh_Min_10_30 = apply(BF_matrix_mineral_10_30, 1, FUN = mean)

**Mixed forest**

MF_matrix = matrix(nrow=nrow(MF), ncol=4)

MF_matrix_mineral = matrix(nrow=nrow(MF), ncol=4)

MF_matrix_mineral_10_30 = matrix(nrow=nrow(MF), ncol=4)

for (i in 1:nrow(MF)) {

  for (j in 1:4) {
```

```
MF_matrix[i,j] =

  # Forest floor litter

  (((MF_Lit_T_b0 * exp(MF_Lit_T_b1 * MF$T_5cm[i])) * # Modell Rh with Temp

  MF_C_Lit * # Multiply by C stocks

  ((MF_M_b0 + MF_M_b1 * MF$VWC_5cm[i] + MF_M_b2 * MF$VWC_5cm[i] ^ 2) / # Correct for
field moisture

  (MF_M_b0 + MF_M_b1 * MF_M_Inc_Lit + MF_M_b2 * MF_M_Inc_Lit ^ 2)))

  +

  # Mineral soil 0-10 cm depth

  ((MF_Min_T_b0 * exp(MF_Min_T_b1 * MF$T_5cm[i])) *

  MF_C_Min_0_10[j] *

  ((MF_M_b0 + MF_M_b1 * MF$VWC_5cm[i] + MF_M_b2 * MF$VWC_5cm[i] ^ 2) /

  (MF_M_b0 + MF_M_b1 * MF_M_Inc_Min + MF_M_b2 * MF_M_Inc_Min ^ 2)))

  +

  # Mineral soil 10-30 cm depth

  ((MF_Min_T_b0 * exp(MF_Min_T_b1 * MF$T_20cm[i])) *
```

MF_C_Min_10_30[j] *

((MF_M_b0 + MF_M_b1 * MF$VWC_20cm[i] + MF_M_b2 * MF$VWC_20cm[i]^2) /

(MF_M_b0 + MF_M_b1 * MF_M_Inc_Min + MF_M_b2 * MF_M_Inc_Min ^ 2))))

MF_matrix_mineral[i,j] =

  # Mineral soil 0-10 cm depth

  (((MF_Min_T_b0 * exp(MF_Min_T_b1 * MF$T_5cm[i])) *

  MF_C_Min_0_10[j] *

  ((MF_M_b0 + MF_M_b1 * MF$VWC_5cm[i] + MF_M_b2 * MF$VWC_5cm[i] ^ 2) /

  (MF_M_b0 + MF_M_b1 * MF_M_Inc_Min + MF_M_b2 * MF_M_Inc_Min ^ 2)))

  +

  # Mineral soil 10-30 cm depth

  ((MF_Min_T_b0 * exp(MF_Min_T_b1 * MF$T_20cm[i])) *

  MF_C_Min_10_30[j] *

  ((MF_M_b0 + MF_M_b1 * MF$VWC_20cm[i] + MF_M_b2 * MF$VWC_20cm[i]^2) /

```
        (MF_M_b0 + MF_M_b1 * MF_M_Inc_Min + MF_M_b2 * MF_M_Inc_Min ^ 2))))

      MF_matrix_mineral_10_30[i,j] =

      # Mineral soil 10-30 cm depth

      ((MF_Min_T_b0 * exp(MF_Min_T_b1 * MF$T_20cm[i])) *

      MF_C_Min_10_30[j] *

      ((MF_M_b0 + MF_M_b1 * MF$VWC_20cm[i] + MF_M_b2 * MF$VWC_20cm[i]^2) /

      (MF_M_b0 + MF_M_b1 * MF_M_Inc_Min + MF_M_b2 * MF_M_Inc_Min ^ 2)))

    }
      }

**Calculate mean values in µmol CO2 kgC^-1 sec^-1**

MF$Rh = apply(MF_matrix, 1, FUN = mean)

MF$RhSE = apply(MF_matrix, 1, FUN = std.error)

MF$Rh_Min = apply(MF_matrix_mineral, 1, FUN = mean)

MF$Rh_Min_10_30 = apply(MF_matrix_mineral_10_30, 1, FUN = mean)
```

---

## Author Comment (AC9) · 26 Nov 2016

The revised manuscript is attached as supplement. All major changes are marked in blue color.

Please also note the supplement to this comment:
http://www.biogeosciences-discuss.net/bg-2016-291/bg-2016-291-AC9-supplement.pdf

---

## Author Response (AR2)

General comments

This manuscript describes measurements of soil respiration (Rs) made in the field in two forests in Bhutan, supplemented by laboratory incubations, that looks at Rs fluxes, sources, seasonally-driven changes, as well as sensitivities to temperature and moisture. This is interesting and valuable, given the paucity of data from this region, if not groundbreaking. The ms is reasonably well written, although there are many minor English errors, and frequently insightful. I particularly liked the comparison between different techniques and measurements/models, even though this is not developed as fully as it could be.
There are some problems. Aspects of the methods and results are unclear; in particular, the authors should be careful to distinguish between measured and modeled results, why and when each was performed, and when they're referring to each. I think the trenched plot results could be better discussed, particularly as the problems that occurred seem relatively straightforward to explain. Finally, and very importantly, the authors need in my opinion to post or include the data and code backing all their main results (see #9 below). In summary, this ms needs moderate to significant revisions in many places, but it's fundamentally a strong and interesting study.

*We thank the anonymous referee for the constructive comments and suggestions. We revised the manuscript accordingly and incorporated all the comments and suggestions. The model code is also included as supplementary files to the manuscript as well as the data.*

Specific comments

1.      Lines 23-24: why does a discrepancy between modeled and measured Ra indicate trenching performed poorly? Clarify. Might also add ", probably because of the short time lag between trenching and measurement"?

*AR: We clarified that and added more discussion about the strengths and weaknesses of both methods (L259-303).*

2.  L. 27: "preceding"?

**AR**: *Changed to "prevailing"*

3.  L. 37: maybe "potentially feeds back on global climate change". Also there are better citations for this, e.g. Frey et al. 2013 (10.1038/nclimate1796) or Wang et al. 2014 (10.1111/gcb.12620)

*AR: Changed and suggested citations added.*

4.  L. 50: start a new paragraph

 *AR: New paragraph set.*

5.  L. 67: "would show decreases in Rs during"

*AR: Changed*

6.  L. 118-: what was delay between trenching and starting measurements?

*AR:Plots were trenched in April 2014. The delay was 1 year accordingly. We stressed this in the revised manuscript (L80).*

7.  L. 172: better to say "Effects of site" rather than "forest type" since you can't actually test forest type (as n=1)

*AR:Changed as "Effects of site" and we consistently use this term in the revised manuscript.*

8.  L. 190: perhaps "to calculate a projected daily field Rs" for clarity

*AR: Re-worded as suggested.*

9. Availability of code and data? It's 2016, and I expect all code and data (at least that backing the main results) to be included as supplementary info, or posted in a repository. It's not acceptable to produce results from a black box

*AR: We have included the R-code as supplementary file and we will also upload the data to a repository.*

10. L. 252-257: interesting!

*AR:Thank you*

11. L. 278: "any meaningful Rh values"?

*Trenching values from additional (2015) trenching plots were taken out (as suggested by reviewers 2 and 3) and discussion was adapted accordingly.*

12. L. 310-: "Q10 tends"; also should be "increase with decreasing temperatures"?

*This discussion was taken out as we avoided Q10 for field Rs (suggested by reviewer 2 and 3).*

13. L. 314: "than the ones"

*Corrected*

14. L. 327: "We intend to"

*Corrected*

15. L. 333: this is awkward English and unclear – why "ambivalent"?

*Deleted.*

16. L. 338: "Rs, falling well within"

*Re-worded.*

17. L. 345: "albeit"? What?

*"besides" was the correct term*

18. L. 348-353: interesting though unsurprising. Might mention this in abstract

*In our opinion this is a little too specific for the abstract.*

19. Figure 1: minor point but perhaps format x axis dates as "Apr 2014", "May 2014", or something like that to eliminate M/D/Y ambiguity (i.e. make consistent with Figures 4 and 5)

*Incorporated in the final revised figures. Labels are consistent as we have removed the data for the year 2014 completely (suggested by reviewers 2 and 3).*

20. Figure 3: necessary?

*We removed the graph.*

21. Figure 4: this is confusing. At the very least, clarify the caption, and perhaps re-think how these data are displayed

*We revised the graph and simplified the caption and data display.*

22. Figure 5: perhaps note in caption that the Rh lines are cumulative

*We improved the caption, it should be clear now.*

**Anonymous Referee #2**

The paper "Soil CO2 efflux from two mountain forests in the Eastern Himalayas Bhutan: components and controls" by Wangdi et al. provides further information on a data poor environment, relevant to the Earths carbon budget. In this way the paper is a useful contribution to the cannon. Further lab-based incubations also appear useful for constraining modeled behaviors in the field, and the provided comparisons to in situ outcomes may be informative.

I have a number of concerns about the choice to include some of the provided data, as well as the exact equations and parameters that the model utilized to determine partitioning between respiration components. Overall, in agreement with the first reviewer, I believe that this study has some sound and useful information and analysis that should be published, but I would expect major revisions would be necessary to clear out the unnecessary components of the article and clarify others.

We highly appreciate all your constructive comments. All suggestions were taken into account and the revised manuscript is shorter and streamlined. 2014 data, which held some methodological bias were removed as suggested by reviewer 2 and 3, Rh-model R-code and all data are added as supplement. While revising the manuscript, we found a minor conversion error for Rh values per kgC-1, which we corrected for in the revised manuscript. Corrections did not affect the overall outcome of the study. Corrections resulted in slightly higher modeled Rh and minimal, insignificant deviations of Q10 and R10 values when compared to the values in the initial manuscript (without any effect on the study results!).

My broad concerns will be listed first with specific items listed afterwards:

1) The "coniferous forest" as described has a substantial component of broadleaved trees ($\sim$29% Quercus sp.), and is later described as, perhaps more appropriately, a "cool temperate mixed coniferous forest" (line 72/73). Perhaps describing as a "mixed forest" throughout the paper would be more helpful. This is likely to have some impact on respiratory fluxes (through litter quality, leaf economics, etc) and this, along with the potential impacts from soil type and understory plant types, density and behavior is not addressed in the text sufficiently in my mind.

We consistently use the term "mixed forest" in the revised manuscript. We discuss possible effects of litter dynamics on Rs and Rh (L250-252, L270-275)

2) I question the value of the 2014 field-based Rs results, considering that they are acknowledged by the authors to be influenced by pressure effects from chamber placement.

We have removed all 2014 data, as suggested. We did not lose important information by doing so, but it made the whole paper much easier to read.

3) I am uncomfortable with different mathematical functions being used to determine the same biological functionality (in particular the linear versus Gaussian response of soil water content to respiration rates). I would prefer that whichever function is used that there is some biological rationale that can be used to defend this choice.

That was a mistake. We use consistent functions in the revised manuscript. We also changed to a more simple mathematical function for the relationship between Rh and soil moisture.

4) I agree with the first reviewer that it would be better to have the model by which Rh and Ra components were calculated either explained through the primary equations in the text, or by incorporating the model as a supplementary material.

We explained the functions in the text more thoroughly and added the R-code of the Rh model as well as all data as supplement.

5) The use of the term Q10 to describe the entire soil response to, effectively, seasonal changes is inappropriate to my mind. By definition Q10 refers to the change in reaction rate of an enzyme or system to 10 degree changes in temperature, and on this basis the lab-based incubation Q10s are appropriate and should be retained and used in the models, but calling the whole system response a Q10 when the authors acknowledge (lines 301-303) that it incorporates water content, leaf litter availability and other co-variable parameters makes this use of the Q10 term meaningless.

We agree that the relationship between seasonal Rs and seasonal soil temperature does not resemble the actual temperature sensitivity of Rs. We also agree that the use of Q10 actually is not desirable in this regard (although quite commonly done). We used another formulation for the exponential relationship in Eq. (1), avoiding the term Q10 already in the Rs function. We completely reworded the results and discussion section and removed the Q10 values for seasonal Rs from the graphs. We did not lose any relevant information by applying these changes, but shortened the ms and made it clearer and easier to read.

1) Line 25/26: see broader point 5 above. This is not in any way a Q10 with the number of conflating variables. Please use different terminology.

No Q10 used for Rs any more – see above.

2) Lines 64-67: These hypotheses are not all that useful and the final hypothesis is not addressed within the paper, leading to a question of whether these are needed in the paper at all.

We removed the hypotheses and defined broader research questions instead.

3) Line 78: Acer campbelli is listed as a dominant species in the cool, temperate mixed coniferous forest but is not listed in Table 1.

We removed *Acer campelli* from the text as it was not dominant.

4) Lines 88-92: Climate can vary dramatically in mountainous regions over spatial scales of 1km. Is there evidence that these weather stations were recording appropriate data for these sites?

Especially rainfall can vary within small spatial scales. However, the climate station was at exactly the same altitude in the same valley/same slope/ same aspect, so that there is no indication of differences in climate at such fine scales. Soil moisture data at the site correspondingly fits very well with rainfall events measured at the weather station.

5) Line 123: By the nature of its close follow on after trenching this seems to refer to volumetric soil water measurements in the trenched plots but instead refers to the broader study plots (as shown in Figure 1). This could be more clear.

We clarified that. Moisture was measured at all plots, broader study plots, trenched plots, and control plots.

6) Lines 145-146: I would be interested in hearing more about the ventilation system used for the incubations. I am uncertain how much water might be lost by the soils during this process (e.g.- the ventilation process during the two-week waiting periods between soil moisture sampling) and how this water loss was addressed during periods between measurements.

We added (L107-109): "In the meanwhile, disconnected containers were ventilated by means of an air pump in order to prevent internal CO2 enrichment. Wet tissues were put into containers in order to prevent samples from drying out during incubations; moisture loss was thereby negligible (< 2 vol. %)".

We also clarified that soil cores were only placed in the incubation chambers during actual incubation runs and that cores were stored in a storage room (+4°C) in-between the incubation runs.

7) Lines 143-148: I wonder about the effect of sieving on Rh considering the disruption placed on the soil/fungal community. It seems likely that this has significantly affected this component within this aspect of the study. (Datta et al Int. Agrophys., 2014, 28, 119-124)

We are aware that sieving disrupts fungal hyphea and has further unwanted effects, such as a disruption of soil aggregates, which could liberate bound SOM. We nevertheless decided sieving the soil. Incubating undisturbed cores, makes it difficult to be sure that root respiration is really excluded, as intact fine roots in the cores can maintain respiration for relatively long times. Correspondingly, root respiration could add to the core CO2 efflux even after long equilibration times and thereby affect the temperature sensitivity. As we aimed to model Rh, we decided not to use intact soil cores. We added some lines to the discussion:

L 280-282: As a last point, soil sieving could have positively affected Rh rates during incubation by releasing physically protected SOM and/or providing additional C sources via disrupted fungal hyphe and fine root fragments (Datta et al., 2014) .

8) Agreed with reviewer #1 point 7

Changed accordingly throughout the revised manuscript.

9) Lines 188-194:  The assumption that the temperature in the soil at 5cm depth is sufficiently predictive of Rs may work within this model but it assumes that the system is sufficiently co-variant that this one data point is essentially all that is needed. This seems to assume that the basal respiration from lower soil depths is effectively constant. Can the authors provide any evidence that this is true?

Most of Rs will be produced in the topsoil with highest SOM, microbial biomass, and fine root contents. Therefore topsoil temperature usually is a quite good predictor of total Rs. We actually do not assume that the basal respiration from deeper layers is constant, but that respiration rates from deeper layers co-varies with that from topsoil. We are aware that this very likely is not really the case as deeper soil temperature reacts somewhat delayed to topsoil temperature variations and as Rs, produced in deep soil, needs some time to diffuse to the surface. The very tight relationship between topsoil temperature and Rs however indicates that deeper soil Rs production is quantitatively not so relevant, or that deeper soil Rs still co-varies with topsoil Rs, at a scale that is mostly covered within the model. We discuss the shortcomings of the model approach in the revised manuscript (L259-284).

10) Line 200:  I agree that a Gaussian distribution is probably the most appropriate here (and for appropriate biologically relevant reasons) but the linear fits later in figure 2 have no real biological rationale.

We changed to a more simple polynomial function which is consistently used now.

11) Lines 205-212: The trenching experiment not only affects water retention in the soil but also provides further litter availability and there are likely non-linear effects that are not well addressed in this section.  I am also unconvinced that the correction for soil moisture is precise and accurate based upon the data reported.  Perhaps it would be more useful to report a range of possible outcomes instead of the firm values reported here.

We more intensively discuss the problems associated with trenching (L285-303). Soil moisture correction should have been fine as we used the moisture measurements directly obtained from trenching and control plots. This was somewhat unclear in the original ms and has been clarified.

12) Line 220:  It is unclear if the lack of specific moisture response function is due to a lack of (or no) collected data or a poor linear or Gaussian fit was obtained from the collected data.

This was matter of a misleading formulation. We simply did not obtain litter $CO_2$ efflux data under different moisture levels. Accordingly, no response function was available for litter, and the function for mineral soil was used instead. We clarified that in the text.

13) Line 246-247: The method for assessing fine root biomass is not reported. Either the method should be discussed or a reference to the data would be helpful.

We added the method (L66-70): Fine root (≤ 2 mm) biomass was estimated by soil-core method (Makkonen and Helmisaari, 1999) once in spring 2014 at both sites. We used a cylindrical soil corer (7.5 cm diameter) for sampling. The extracted core samples were divided into three depth sections of 0-10 cm, 10-20 cm and 20-30 cm. After washing and sorting (live roots and necromass), roots were dried at 70 °C to constant mass before weighing dry biomass. Contribution of fine root C was estimated as 50 % of the plant tissue.

14) Lines 252-254: Given the potentially compromised nature of the Rs data from 2014 I would prefer that it not be reported at all, especially given the successful campaign run through 2015.  The nature of pressure pumping and its effects on fluxes is sufficiently well established that this doesn't add much value to the paper.

We removed all 2014 data as suggested.

15) Line 259:  This is somewhat self-fulfilling.  You measured once every three weeks and find that 3 week sampling density is sufficient. In order to truly test this you would need a higher density sampling rate that you are then able to sub-sample at the 3 week frequency. I would suggest this comment (and others similar) be removed from the text.

This is true. We removed that.

16) Line 277-278: Again, given the nature of the trenched plots in 2015 (errors in strategy that are explainable and understandable) I am uncertain why this is discussed in the methods section and here. If I understand correctly not including this would save space and would not affect your analysis.

We removed the additional 2015 trench plot data completely.

17) Lines 280-287: The model should be made clearer, in agreement with point 9 from reviewer #1.

We added the complete code and data as supplement.

18) Lines 301-319: I find this justification of the "field Q10" values to be unconvincing and suggest that this section be reworked or removed from the text. There are too many other variables that are not addressed beyond the already tenuous soil moisture correction for this to be adequately compared to a true Q10.

We re-worked this chapter. Actually, we deleted most of it as we decided not to use Q10 for Rs (as suggested). The whole chapter now is much clearer and only the important information is provided/discussed.

I agree that Figure 3 seems to serve little purpose and any lost detail can be described quickly and easily in the text.

We removed the graph.

**Anonymous Referee #3**

This manuscript entitled "Soil CO2 efflux from two mountain forests in the Eastern Himalayas Bhutan: components and controls" by Wangdi et al. provides interesting, relevant and valuable information on a poorly studied region. Manuscript is mostly well written and easy to read. The use and comparison of different techniques of measurements and different models is interesting. Nevertheless, aspects of the methods and then of the results remained unclear because it was not easy to distinguish and understand when and also why measured or modeled results were used to suit the purpose.

Major revisions would be necessary to clarify the manuscript and to develop more explicitly the objectives of the comparison between different measurements/models.

We highly appreciate the constructive comments which largely aligned with suggestions from reviewers 1 and 2. We better distinguish between modeled Rh and trenching results in the revised manuscript and we better explained why these methods were used. All other suggestions have been followed as well. While revising the manuscript, we found a minor conversion error for Rh values per kgC-1, which we corrected for in the revised manuscript. Corrections did not affect the overall outcome of the study. Corrections resulted in slightly higher modeled Rh and minimal, insignificant deviations of Q10 and R10 values when compared to the values in the initial manuscript (without any effect on the study results!).

General comments:

I am not convinced that 2014 field Rs data should be presented in the ms as they are not relevant because influenced by pressure effects. In the same way given the trenched plots in 2015 didn't produced meaningful values, what does these data bring to the analysis? If retained, the trenched plot results could be better discussed. Important care must be given to distinguish between measured and modeled results. Authors should explain why and when each was performed and also why and when they are referring to each.

As suggested, we removed the 2014 Rs data as well as the data from 2015 trenching plots. The text is now easier to read and we did not lose relevant information.

Specific comments:

1) l.23-24 : unclear. I can't see why the variability of Ra indicates a methodological issue with the trenching

We removed this from the abstract. We further refined the discussion and point at shortcomings of both methods (model, trenching) in the discussion section (L259-303).

2) l.272 : prefer effect of sites rather than of forest type

Changed to "sites" in the whole manuscript

3) l.190 : discuss how constraining the model with the temperature in the soil at 5cm depth is sufficient and relevant. What about the deepest contributions to Rs?

We actually used soil temperatures from different soil layers (mineral soil 5 cm, and mineral soil 20 cm depth) for modeling (L180-183) of $CO_2$ efflux from the corresponding layers. $CO_2$ efflux from < 30 cm depth was neglected in our model. We discussed this in terms of the Rh model outcome. We extended the discussion accordingly (L259-284).

4)l.190 :  The same parameters (of Eq1) are used to model Rs over the year without any discussion whether or not the Q10 could vary with the temperature range over the year.

We alternatively fitted a Gaussian function (where Q10 changes with temperature). The fit of the simple exponential function was slightly better. We therefore decided to stick to this function.

5)l.205-212 :  agreed with reviewer #2 point 11.  Indicate the uncertainties rather than that corrected value.

As both methods have some different sources of uncertainty, they are quite difficult to quantify. We therefore stuck to the graph, but better discuss uncertainties in the text.

6) l.218: what is Fig S1 ?

Supporting Figure 1 is a supplement. We deleted this figure as this is a common procedure in model anyway.

7) l.246 : report and discuss the method used to estimate fine root biomass

We added the method (L66-70).

8) l.259 : How can you be convinced that it 'indicates that a three-week interval is sufficient' although you didn't measured with a higher frequency ? Restrain the purpose.

That's true. We deleted this sentence.

9) l.278 : useful ?

Deleted.

10) l.308-319: I have issues with the analysis presented here because I am concerned about the definition for the terms intrinsic and apparent sensitivities.  Recently, Sierra et al. 2015, JAMES 7: 335-356 proposed consistent and formal definitions for intrinsic and apparent sensitivity.  It would be nice if the authors referred to that definition or explained how they defined these conceptual sensitivities.

We completely revised the section. According to reviewer 2, we don't use Q10 for field Rs. The text passage is shorter and much easier to read now. We refer to Sierra et.al. in the discussion section with regard to moisture effects on temp sensitivity.

11) l.345: albeit ?

Should be "besides" – changed.

12) Figure 4: The figure is really confusing. Caption doesn't help

We adapted the figure. Should be easy to understand now.

13) Figure 5:  not easy to understand that the lines are cumulative.  Indicate by filling with different colors that the bottom area is Rh (10 – 30), the second area is Rh (0 –10), the third (upper) one Rh litter and the highest Ra.

We adapted the figure and caption. It should be clear now.

**4. Associate Editor's Comments**

**General Comments**

Many thanks for your revised manuscript. You have managed to address the referees' comments well, and I'm happy in principle for this study to be published. I am also glad to see that you have made your source data available via figshare. Could I kindly ask you to address a number of remaining, mostly minor points that I list below? These are mainly clarifications in the text.

**Specific Comments**

Abstract: Please include a real reference when quoting annual Rs figures, i.e. "14.5 ± 1.2 t C ha-1 broadleaved; 12.8 ± 1.0 t C ha-1".

**AR**: Added.

Line 6: Either "… serve as a C sink or source…", or "… serve as C sinks or sources…".

**AR**: Changed to "serve as C sinks or sources".

Line 61: Should this be "0-10, 10-20 and 20-30 cm"?

**AR:** Sorry, yes, it was a typo. It should be "0-10, 10-20 and 20-30 cm"

Line 184: This use of input as a proxy for stock is unclear. Do you have data to back this up? (see also comment below).

**AR:** Due to a strong seasonal variation in litter input/decomposition (at both sites, but especially at the broadleaved site), litter layer depth and C stocks will vary with time. We think that thoroughly measured total annual litter input rates represent a reasonable estimation for this year's litter stocks and is more accurate than a single point in time measurement of the litter layer depth/stock. A thorough assessment of litter C stocks (multiple-time assessment throughout the seasons) was not carried out during this study. We therefore decided to stick to the litter input as proxy. Nevertheless, a rough estimate during spring 2015 showed that stocks roughly matched annual 2015 litter input rates. We added this to the method text and better explained the effects on uncertainty in the discussion (see below).

Methods (L183-187): We used the litter $Q_{10}$ together with continuous temperature at 5 cm soil depth to model daily Rh from the litter layer. In order to scale to field fluxes, we used the annual litter input (Table 1) as a proxy for field litter C stocks. A first rough litter assessment in March 2015 showed that litter stocks were in a similar range as the annual litter input at both sites.

Line 258: I'm not sure the word "disabled" works here. Stating that belowground allocation of C by trees is limited should be appropriate to make this point.

**AR:** We removed the word "disabled".

Line 260: Better: "However, there is…"

**AR**: Corrected.

Line 269: This is unclear to me. In steady state, litter C input is balanced by litter C losses, but the pool size of litter is not necessarily equal to one year's amount of litter. That depends on turnover rates. Please clarify.

**AR:** We agree, that annual litter input is not necessarily equal to the actual amount of litter on the forest floor. However, both, litter input rates and litter decomposition rates underlie a strong seasonal variation (especially in the broadleaved forest); depending on the time of the year, an exact quantification of annual litter stocks can thus be quiet challenging (e.g. spring VS. fall measurements of litter layer thickness). We therefore think that total litter input rates as a rough proxy for the C stocks of this year's litter constitutes fair a compromise. We are aware that our upscaling approach of annual heterotrophic respiration rates to the field holds a lot of uncertainties; the estimation of litter C stocks is recognized as one of them (please see discussion section). However, the respiratory contribution from the litter layer to total heterotrophic respiration rates was comparably low at both sites. Although the magnitude of predictions seems to be overestimated, we nevertheless think that our method represents an applicable and easy tool to investigate the seasonal course of heterotrophic and autotrophic soil respiration rates. We changed the discussion to:

L 276-279: Using annual litter input as proxy for litter C stocks is a further source of uncertainty. Litter input has temporal patterns and thereby affects litter decomposition dynamics. Such temporal patterns in litter input/decomposition were not reflected in our model. The modeled contribution of the litter layer to total soil Rh was, however, small (Fig. 4), and therefore, the uncertainty related to temporal litter layer dynamics can also be considered as small.

Line 310: It was either similar or not… Do you mean "almost identical"?

**AR:** Changed to "almost identical".

Lines 315-322: I'm not sure why you give a very approximate soil C budget when stating that you don't actually have the data to do so confidently. This was not your aim, and I don't see that this is a meaningful contribution at this point.

**AR:** We deleted this chapter and moved the root and litter discussion (L317-323) upwards (now L249-254).

Below you find the revised manuscript. All major changes are marked in blue and red .

[revised manuscript text omitted]